# Assessing the impacts of various factors on circular RNA reliability

Trees-Juen Chuang, Tai-Wei Chiang, Chia-Ying Chen

**Circular RNAs (circRNAs) are non-polyadenylated RNAs with a continuous loop structure characterized by a non-colinear back-splice junction (BSJ). Although millions of circRNA candidates have been identified, it remains a major challenge for determining circRNA reliability because of various types of false positives. Here, we systematically assess the impacts of numerous factors related to circRNA identification, conservation, biogenesis, and function on circRNA reliability by comparisons of circRNA expression from mock and the corresponding colinear/ polyadenylated RNA–depleted datasets based on three different RNA treatment approaches. Eight important indicators of circRNA reliability are determined. The relative contribution to variability explained analyses reveal that the relative importance of these factors in affecting circRNA reliability in descending order is the conservation level of circRNA, full-length circular sequences, supporting BSJ read count, both BSJ donor and acceptor splice sites at the same colinear transcript isoforms, both BSJ donor and acceptor splice sites at the annotated exon boundaries, BSJs detected by multiple tools, supporting functional features, and both BSJ donor and acceptor splice sites undergoing alternative splicing. This study thus provides a useful guideline and an important resource for selecting high-confidence circRNAs for further investigations.**

## Introduction

CircRNAs are an emerging class of RNAs formed by *cis*-backsplicing with a covalently closed loop structure that are endogenously expressed as non-polyadenylated circular molecules (1, 2). Such a loop structure is characterized by a non-colinear (NCL) back-splice junction (BSJ) between a downstream splice donor site and upstream splice acceptor site. Despite most circRNAs are expressed at a much lower level compared with linear mRNAs (3, 4, 5), some are highly expressed (3, 6), even more abundant than their colinear counterparts (7, 8). In addition, circRNAs are more stable than their corresponding colinear mRNA isoforms (1, 9). Some circRNAs were

further demonstrated to be evolutionarily conserved across species (3, 10, 11, 12). Accumulating evidence demonstrated that circRNAs can participate in various aspects of biological processes, including competing with their host gene expression and splicing, regulating activities of miRNAs and RNA-binding proteins (RBPs), and acting as translation templates (2, 13). CircRNAs were reported to be especially abundant in brain and neuronal tissues (11, 14) and regulatory important in nervous system development and aging (15). Loss or dysregulation of circRNAs may affect brain function (12, 15, 16, 17), indicating the roles of circRNAs in pathogenesis and progression of neurological diseases. In cancer studies, alternation of circRNA expression was shown to affect cell apoptosis, invasion, migration, and proliferation, revealing the potential of circRNAs for serving as biomarkers and therapeutic targets in cancer (18, 19). These findings suggest that circRNAs are an ancient and fine-tuned class of functional transcripts.

Varying bioinformatic tools have been developed to identify BSJs using high-throughput transcriptome sequencing (RNA-seq) data (20), leading to a tremendous number of potential circRNA candidates in human and other species (21, 22, 23). However, circRNA detections are hampered by various types of false positives. An NCL junction that originates from sequencing error, in vitro artifact, ambiguous alignment, *trans*-splicing, or genetic rearrangement is often misjudged as a BSJ event by computational detections (1, 2). This reflects the fact of limited overlap among the circRNA candidates identified by different tools (1, 24, 25, 26, 27) or databases (23), highlighting the issue of circRNA reliability. Although millions of circRNA candidates were deposited in varied databases, only a very limited number of candidates have been carefully curated to date (23). It requires a guideline for evaluating the reliability of the detected circRNA candidates before performing time- and cost-consuming experimental validations.

Because circRNAs are NCL RNAs and lack 3′polyadenylated tails, many circRNA candidates were detected in RNA-seq datasets derived from total RNAs (the ribosomal RNA–depleted RNAs without poly(A)-selection; "mock RNAs" for simplicity), which contain RNA-seq reads from both colinear and NCL RNAs (28). Because most colinear RNAs are digested by RNase R (3, 28, 29, 30), a detected circRNA candidate is more likely to be real if its normalized BSJ read count is enriched or not reduced after the treatment. Therefore, comparisons of BSJ read counts detected from mock and RNase

---

Genomics Research Center, Academia Sinica, Taipei, Taiwan

Correspondence: trees@gate.sinica.edu.tw.

R–treated RNA datasets were often employed to evaluate the reliability of the detected circRNAs (24, 25, 26, 31). However, in some cases, such a mock-treated comparison approach may not be efficient for circRNA detection because of two inherent limitations. First, some colinear RNAs containing highly structured 3′ ends or G-quadruplexes are also resistant to RNase R (32). Second, some circRNAs may be susceptible to RNase R and digested by this exonuclease (33). Accordingly, in addition to RNase R–treated RNAs, we considered two other RNA treatments, RNAs treated with coupling A-tailing and RNase R digestion and RNAs treated with depletion of both ribosomal RNAs and polyadenylated RNAs, for mock-treated comparisons. The former could efficiently digest most of the RNase R–resistant colinear RNAs stated above (32), and the latter could deplete most polyadenylated RNAs and retain circRNAs susceptible to RNase R (33). On the basis of comparisons of normalized BSJ read counts detected from mock and the corresponding colinear/poly(A)-depleted RNA datasets from three different RNA treatments, we thus presented a systematical workflow to evaluate the impacts of various factors related to circRNA identification, conservation, biogenesis, and function on circRNA reliability. After determining statistically important indicators of circRNA reliability, we further assessed the relative influence of these important indicators on circRNA reliability. Collectively, we provided a useful guideline for selecting high-confidence circRNAs. All of the information related to circRNA identification, conservation, biogenesis, and function was freely available.

# Results

## Potential false positives in the circRNA database

As shown in our analysis flow (Fig 1A), we first extracted circRNA (or BSJ) candidates from the circAtlas database 2.0 (22) (580,654 candidates; see also Table S1). Like previous reports (1, 24, 25, 26, 27, 34), a considerable percentage (36.5%) of the examined circRNA candidates were detected by only one tool (Fig 1B). Such great discrepancies in the identification results among tools reflected the uncertainty of the computationally detected circRNA events (1, 24, 25, 26, 27). One cause of the false-positive calls may be due to alignment ambiguity with an alternative colinear explanation or multiple hits when detecting BSJs (1, 26, 35). Of the 580,654 circAtlas circRNAs, we found that 17.3% (100,183 circRNAs; Fig 1C) were potential false positives arising from an alternative colinear explanation (Fig S1) or multiple hits (see the Materials and Methods section). It was reported that the circRNAs detected by multiple tools tended to be more reliable than tool-specific circRNAs (36, 37). As expected, the percentages of circRNAs derived from ambiguous alignments significantly decreased with increasing the numbers of the detected tools (Fig 1D), reflecting the positive correlation between the level of circRNA reliability and the number of circRNA detection tools. For accuracy, we excluded the circRNA candidates arising from potential alignment ambiguity (100,183 events) and only considered the remaining circRNAs (480,471 events) for the following analyses of circRNA reliability.

Most circRNA candidates were detected from RNA-seq datasets derived from mock samples (the ribosomal RNA–depleted RNAs without poly(A)-selection), which contain RNA-seq reads from both circRNAs and colinear/polyadenylated RNAs (28). We employed comparisons of normalized BSJ read counts detected from mock and the corresponding colinear/poly(A)-depleted datasets to determine high-confidence circRNAs. For comprehensively assessing circRNA reliability and increasing the robustness of our analyses, we extracted 19 mock-treated sample pairs of different samples or studies from public datasets, which included three types of RNA treatment approaches: RNase R approach (group 1), A-tailing RNase R approach (group 2), and non-poly(A) approach (group 3) (see Tables 1 and S2). We examined the 480,471 circRNAs in all the mock samples (see the Materials and Methods section) and found that a considerable percentage of circRNAs (38–80%) were supported by only one BSJ read (Fig 1E). To minimize potential false positives, only the circRNAs expressed in the mock samples with the supporting BSJ read count ≥ 2 were considered in the analysis for each mock-treated sample pair. It was presumed that the circRNAs not depleted after the treatment ("not-depleted circRNAs") were more likely to be bona fide circRNAs than those depleted because of the treatment ("depleted circRNAs"). We then compared not-depleted circRNAs with depleted ones in terms of various factors related to circRNA identification, conservation, biogenesis, and function (Table S3), and thereby accessed the importance of these factors in affecting circRNA reliability.

## Factors related to circRNA identification

Regarding the circRNAs identified in the mock samples, the impacts of the factors related to circRNA identification (i.e., BSJ read count, number of detected tools, and evidence of full-length circular sequence) on circRNA reliability were assessed. First, we found that the supporting BSJ read counts were significantly higher in not-depleted circRNA candidates than in depleted ones in all mock-treated sample pairs examined (all FDR < 0.05; Fig 2A). The percentages of non-depleted circRNAs increased with increasing the supporting BSJ read counts (Fig 2A). Second, we examined the influences of BSJs detected by multiple or single circRNA detection tools on circRNA reliability and found that the former were significantly enriched for not-depleted circRNAs (Fig 2B). Generally, the percentages of non-depleted circRNAs increased with increasing numbers of the detected tools (Fig 2B). Third, we examined the influences of BSJs with or without the evidence of full-length circular sequences reconstructed by CIRI-full (43) on circRNA reliability and showed that the former were significantly enriched for not-depleted circRNAs, regardless of the RNA treatment approaches (Fig 2C). Because the performance of CIRI-full, a short read–based approach, for identifying full-length sequences of circRNAs is hampered by the length of short reads (43), we also extracted the full-length circRNA candidates identified by circFL-seq based on nanopore long reads (44). Such a long read–based approach provided the direct support of full-length circular sequences from single-molecule long reads. Similarly, the BSJ events with the evidence of full-length circular sequences showed a significant enrichment for not-depleted circRNAs in all mock-treated sample pairs examined (Fig 2C).

## Factors related to circRNA conservation

We then examined the relationship between circRNA reliability and conservation patterns at the three levels: species, tissues, and

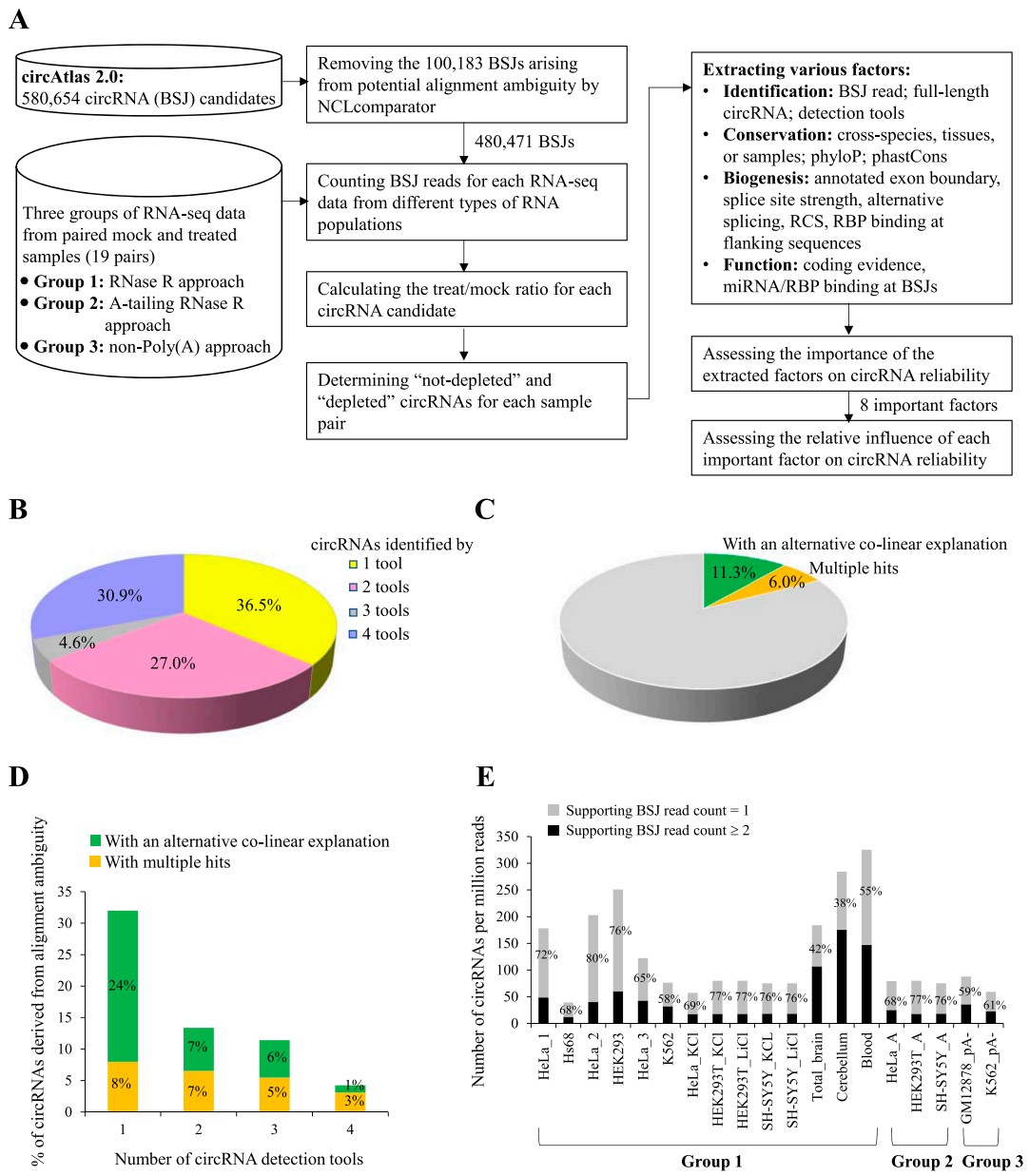

**Figure 1. The assessment of the impacts of various features on circRNA reliability.**
**(A)** Flowchart of the overall analyses. **(B)** Distribution of the extracted circAtlas circRNA (BSJ) candidates (580,654 candidates) identified by 1, 2, 3, or 4 circRNA detection tools. **(C)** 580,654 candidates derived from potential alignment ambiguity (with an alternative colinear explanation or multiple hits). **(D)** Comparisons of the percentages of circRNA candidates derived from potential alignment ambiguity for the candidates detected by 1, 2, 3, or 4 tools. **(E)** Comparisons of normalized numbers of circRNA candidates with supporting BSJ read count = 1 or ≥ 2 in all extracted mock samples of the 19 mock-treated sample pairs. For each sample, the percentage of circRNA candidates supported by one BSJ read is shown.

individuals (or samples). We found that the conservation levels of circRNAs at the three levels (Fig 3A–D) were all significantly higher in not-depleted circRNAs than in depleted ones, regardless of the RNA treatment approaches. The percentages of non-depleted circRNAs increased with increasing numbers of the conserved species (Fig 3A), tissues (Fig 3B), and samples (Fig 3D). Of note, in contrast with the number of tissues (Fig 3B), the levels of tissue specificity index were significantly lower in not-depleted circRNA than in depleted ones (Fig 3C). We further showed that the evolutionary rates (as determined by phyloP (45) or phastCons (46)) of the sequences

around the BSJs (see see the Materials and Methods section) were generally higher in not-depleted circRNAs than in depleted ones (Fig S2). These results suggested that the BSJ events widely detected across species/samples were more reliable than those detected in a limited number of species/samples.

## Factors related to circRNA biogenesis

We proceeded to examine the relationships between circRNA reliability and the factors related to circRNA biogenesis. Because

**Table 1. Summary of RNA-seq data used in this study[a].**

| Approach (mock versus treated RNAs)[b] | Sample pair ID | Resource (ref.) |
|---|---|---|
| Group 1: RNase R approach | | |
| rRNA- versus RNase R+ | HeLa_1 | PRJNA266072 (38) |
| | Hs68 | SRP011042 (3) |
| | HeLa_2, HEK293 | PRJNA266072 (39) |
| | HeLa_3, K562 | PRJNA231721 (40) |
| | Total_brain, cerebellum, and blood | PRJNA646808 (34) |
| rRNA- (KCl) versus RNase R+ (KCl) | HeLa_KCl | PRJNA541935 (32) |
| rRNA- versus RNase R+ (KCl) | HEK293T_KCl and SH-SY5Y_KCl | PRJNA752779 (41) |
| rRNA- versus RNase R+ (LiCl) | HEK293T_LiCl and SH-SY5Y_LiCl | PRJNA752779 (41) |
| Group 2: A-tailing RNase R approach rRNA- (LiCl) versus A-tailing/RNase R+ (LiCl) | HeLa_A | PRJNA541935 (32) |
| rRNA- versus A-tailing/RNase R+ (LiCl) | HEK293T_A and SH-SY5Y_A | PRJNA752779 (41) |
| Group 3: non-poly(A) approach rRNA- versus p(A)- | GM12878_pA- and K562_pA- | ENCODE (42) |

[a]The detailed information is given in Table S2.

[b]rRNA-, RNase R+, A-tailing/RNase R+ (LiCl), and p(A)-represented rRNA-depleted RNAs, rRNA-depleted RNAs with RNase R digestion, rRNA-depleted RNAs treated with coupling A-tailing and RNase R digestion in LiCl-containing buffer, and rRNA-/poly(A)-depleted RNAs.

circRNAs were suggested to be generated by canonical spliceo-somal machinery (1, 5, 47, 48, 49), we examined whether the BSJs of not-depleted circRNA candidates tended to agree to well-annotated exon boundaries of colinear transcripts or be subject to alternative splicing (AS) based on previously annotated colinear transcripts (e.g., the Ensembl annotation) as compared with those of depleted circRNA ones. Indeed, the BSJs that agreed to annotated exon boundaries (either donor or acceptor splice sites) showed a significant enrichment for not-depleted circRNAs, regardless of the RNA treatment approaches (Figs 4A and S3A). The BSJs with both donor and acceptor splice sites matching annotated exon boundaries from the same colinear transcript isoforms were also enriched for not-depleted circRNAs (Fig 4B). We also observed that the splice site strength of BSJs was generally stronger in not-depleted circRNAs than in depleted ones, regardless of the tools used for estimating the splice site strength (Fig S3B). Regarding the BSJs matching previously annotated exon boundaries of colinear transcripts, the BSJs (donor or acceptor splice sites) that were subject to AS were significantly enriched for not-depleted circRNAs across the mock-treated sample pairs examined (Fig S3C). Such a trend exhibited particularly enriched when both donor and ac-ceptor splice sites of the BSJs were subject to AS (Fig 4C). These observations support that most circularized exons are recognized by the spliceosome during canonical splicing.

Next, previous studies demonstrated that backsplicing can be facilitated by RCSs residing in the sequences flanking circularized exons (3, 8, 17, 29) and affected by the competition of RCSs across flanking regions (RCS$_{across}$) or within individual regions (RCS$_{within}$) (29). We thus asked whether the BSJs with RCS$_{across}$ in flanking regions were enriched for not-depleted circRNA candidates. However, we did not observe this trend (Fig 4D). We further examined the numbers of (RCS$_{across}$ - RCS$_{within}$) for the detected circRNAs and found no signif-icant differences between not-depleted and depleted circRNAs in terms of the percentages of the BSJs with (RCS$_{across}$ - RCS$_{within}$) ≥ 1

(Fig 4E). In addition, a number of RBPs were demonstrated to serve as a role for regulating circRNA biogenesis by binding to cis elements in the sequences flanking circularized exons (1, 2). We then examined whether the factor of BSJs with RBPs binding to the flanking regions showed enrichment for not-depleted circRNAs and found that this factor was slightly enriched for not-depleted circRNAs only (Fig 4F). We also examined the BSJs with RCS$_{across}$ or RBP-binding sites in flanking regions and did not observe a significant enrichment for not-depleted circRNAs (Fig 4G). Moreover, considering the minimum distance of the detected RBP-binding sites to the BSJs, the distance was only slightly shorter in not-depleted circRNAs than in depleted ones (Fig 4H). These results suggested that the factors of RCS or RBPs binding to the flanking regions were not good indicators for evaluating circRNA reliability.

## Factors related to circRNA function

We then examined the impact of functional features (miRNA binding, RBP binding, and seven types of evidence for circRNA coding potential) on circRNA reliability (see the Materials and Methods section). For each BSJ event, because the circularized sequence is shared with its colinear counterpart except for the BSJ donor and acceptor splice sites, we con-sidered the predicted miRNA/RBP-binding sites and ORFs spanning the BSJ. Integration of these nine types of functional features, we observed that numbers of supporting functional features were significantly higher in not-depleted circRNAs than in depleted ones in all mock-treated sample pairs (all FDR < 0.05; Fig 5). The percentages of non-depleted circRNAs increased with increasing numbers of supporting functional features (Fig 5). These results suggested that the strength of functional evidence for circRNA candidates were positively correlated with the level of circRNA reliability.

## Relative effect of each individual factor on circRNA reliability

According to the above assessment results, we observed eight important indicators of circRNA reliability, each of which can

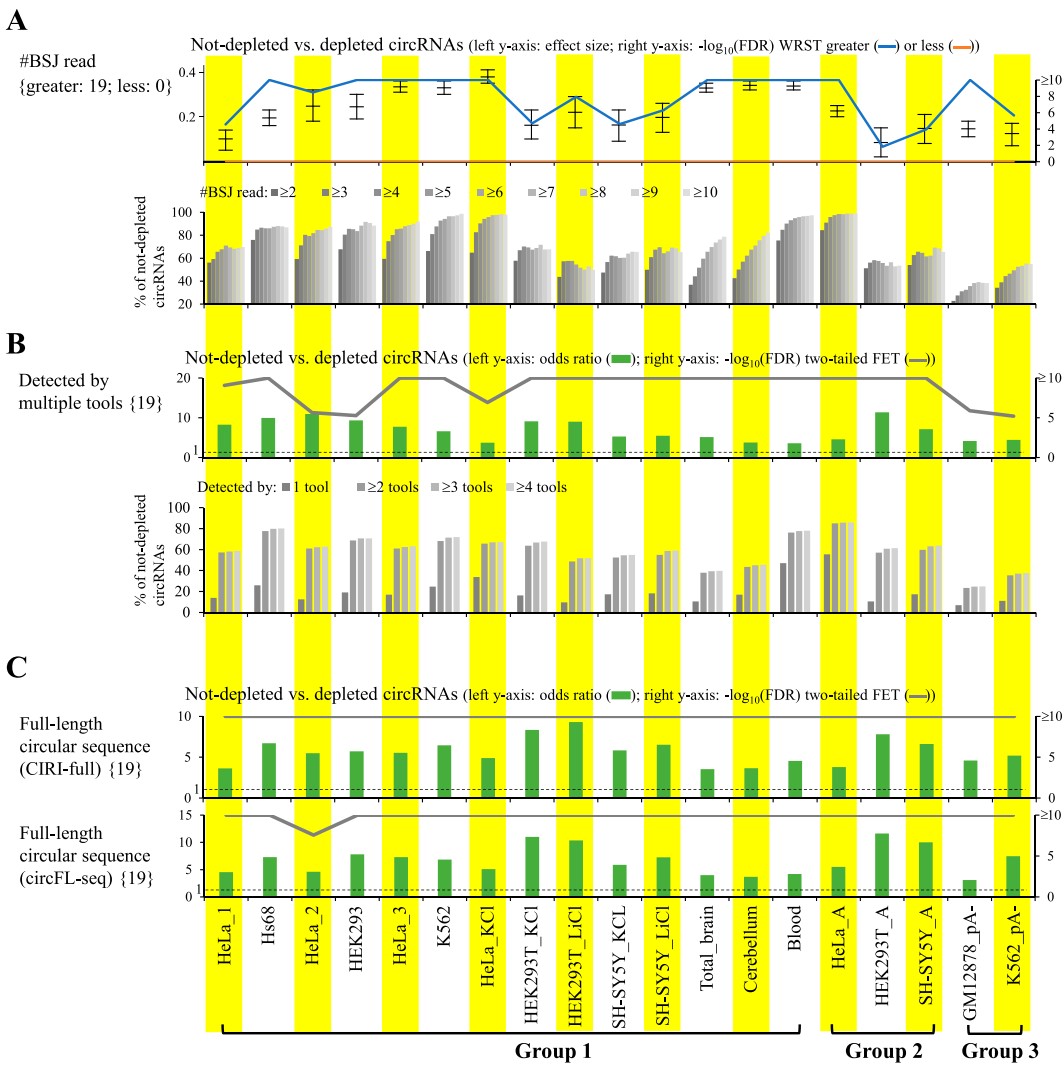

**Figure 2. Impact of factors related to circRNA identification on circRNA reliability.**
**(A)** Supporting BSJ read count, **(B)** number of detected tools, and **(C)** evidence of full-length circular sequence. For (B, C), the odds ratios, which were determined using two-tailed Fisher's exact test, represented the ratios of the occurrences of (B) circRNAs detected by multiple tools or (C) circRNAs with the evidence of full-length circular sequence for non-depleted circRNAs to the occurrences of those for depleted circRNAs. The dashed lines represented odds ratio = 1. **(A, B)** For the bottom panels of (A, B), the correlations between percentages of non-depleted circRNAs and (A) supporting BSJ read count or (B) number of detected tools are shown. For (C), the evidence of full-length circular sequence was supported by CIRI-full (a short read-based approach, top) or circFL-seq (a long read-based approach, bottom). **(A, B, C)** *P*-values were determined using Wilcoxon rank-sum test (WRST; greater or less; (A)) or two-tailed FET (B, C) and FDR adjusted across 19 mock-treated sample pairs for each examined factor using Benjamini–Hochberg correction. For (A), the Wilcoxon effect sizes and the corresponding 95% confidence intervals are plotted (see also Table S4). The number of mock-treated sample pairs that passed the statistical significance tests with FDR < 0.05 (or –log$_{10}$(FDR) > 1.3) are represented in curly brackets.

effectively distinguish between non-depleted and depleted circRNAs in all mock-treated sample pairs examined. The eight factors were (i) supporting BSJ read count; (ii) BSJs detected by multiple tools; (iii) full-length circular sequences; (iv) conservation level of circRNA (number of samples observed the circRNAs); (v) both BSJ donor and acceptor splice sites at the annotated exon boundaries; (vi) both BSJ donor and acceptor splice sites at the same colinear transcript isoforms; (vii) both BSJ donor and acceptor splice sites undergoing annotated AS events; and (viii) supporting functional features. We then employed a generalized linear model (GLM) with the eight factors and measured the relative contributions to variability explained (RCVE) (50) to evaluate the relative effect of individual factor in affecting

circRNA reliability for each mock-treated sample pair (see the Materials and Methods section) (Fig 6A and Table S4). We ranked the RCVE scores for each mock-treated sample pair and found that on average the most important features in descending order were the conservation level of circRNA, full-length circular sequences, supporting BSJ read count, both BSJ donor and acceptor splice sites at the same colinear transcript isoforms, supporting functional features, both BSJ donor and acceptor splice sites at the annotated exon boundaries, BSJs detected by multiple tools, and both BSJ donor and acceptor splice sites undergoing annotated AS events (Fig 6B).

Particularly, our result revealed that the conservation level of circRNA (number of samples observed the circRNAs) is the most

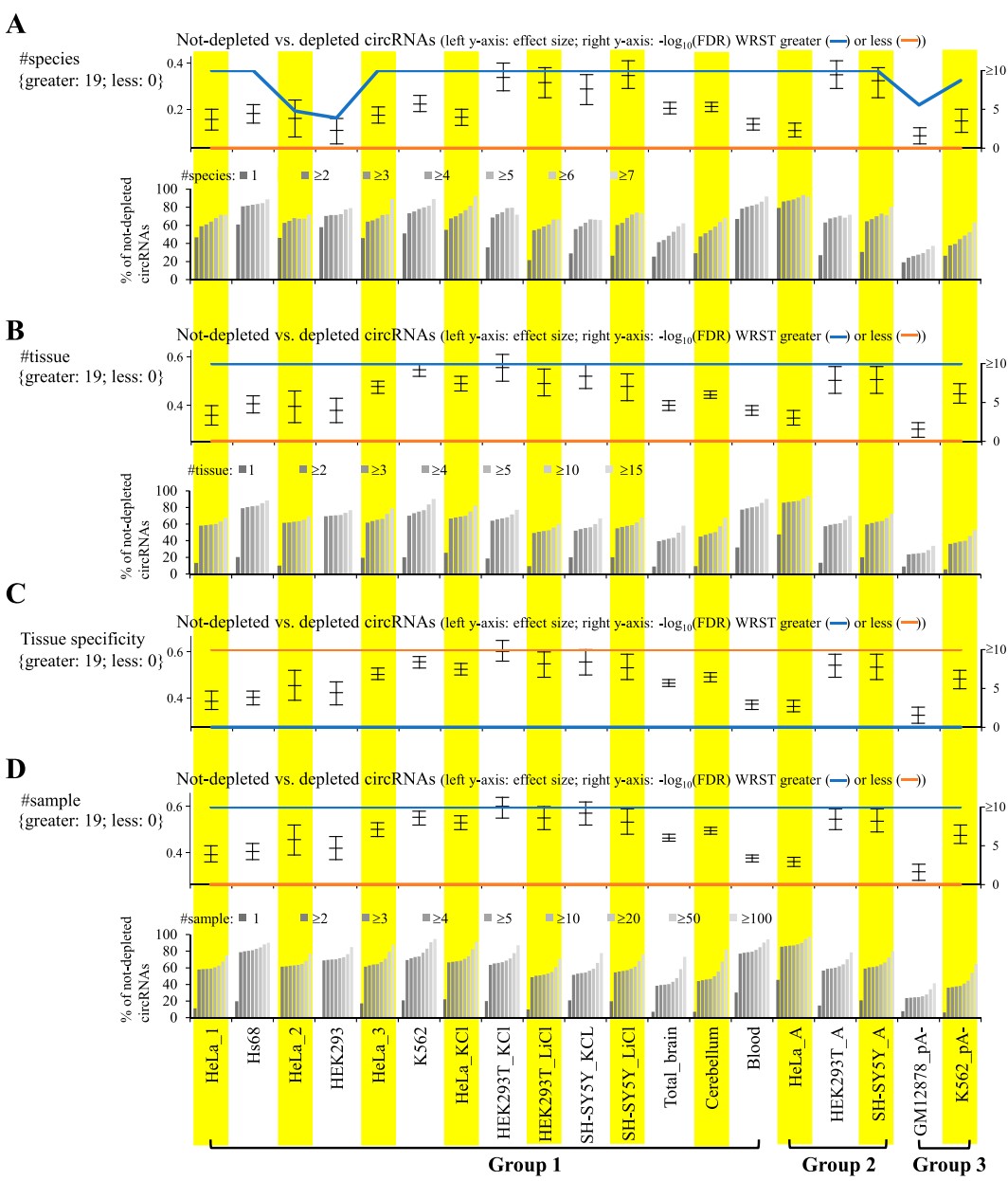

**Figure 3. Impact of factors related to circRNA conservation on circRNA reliability.**
**(A, B, C, D)** Impact of conservation factors at the (A) species, (B, C) tissue, and (D) individual (or sample) levels on circRNA reliability. For the bottom panels of (A, B, and D), the correlations between percentages of non-depleted circRNAs and (A) number of conserved species, (B) number of conserved tissues, or (C) number of conserved samples are shown. *P*-values are determined using WRST (greater or less) and FDR adjusted across 19 mock-treated sample pairs for each examined factor using Benjamini–Hochberg correction. The Wilcoxon effect sizes and the corresponding 95% confidence intervals are plotted (see also Table S4). The number of mock-treated sample pairs that passed the statistical significance tests with FDR < 0.05 (or −log$_{10}$(FDR) > 1.3) are represented in curly brackets.

dominant determinant of circRNA reliability common to all the examined mock-treated sample pairs except for the sample pair in blood (Fig 6B). Of note, the samples considered were the integration of samples across species, tissue, and individuals in circAtlas. It is known that circRNAs are often expressed in a cell type-/tissue-specific manner (4, 51). Actually, we found that most (61%; 293,152 circRNAs) of the 480,471 circAtlas circRNAs were detected in one sample only (Table S3). We were curious about whether the positive correlation between the percentage of not-depleted circRNAs and the number of samples observed the circRNAs held when considering

one cell type/tissue only. Of the examined mock-treated sample pairs, the RNA-seq data of five pairs (HeLa_1, HeLa_2, HeLa_3, HeLa_KCl, and HeLa_A) were all derived from HeLa cells, which were generated from different studies or conditions (Table S2). The five HeLa-based RNA-seq datasets can be regarded as replicates. As expected, the percentage of not-depleted circRNAs was strongly positively correlated with the number of samples observed the circRNAs in all the five HeLa-based pairs (Fig 6C, left). The similar trend was observed for the two K562-based pairs (Fig 6C, middle). We also examined the HeLa_1 and HeLa_2 mock-treated sample

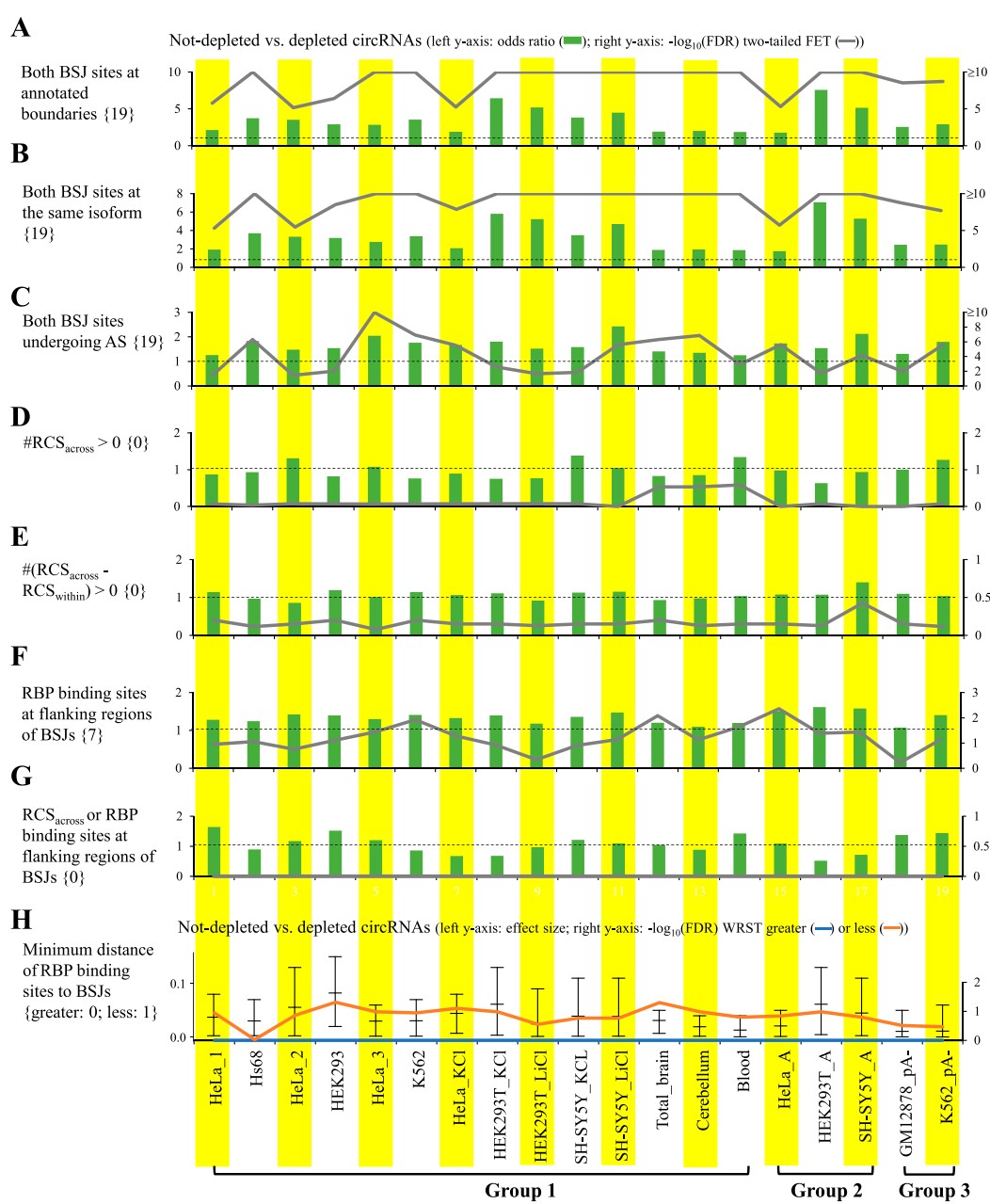

**Figure 4. Impact of factors related to circRNA biogenesis on circRNA reliability.**
**(A)** Both BSJ sites at annotated boundaries, **(B)** both BSJ sites at the same isoform, **(C)** both BSJ sites undergoing alternative splicing (AS), **(D)** BSJs with #$RCS_{across}$ > 0,
**(E)** BSJs with #($RCS_{across}$ - $RCS_{within}$) > 0, **(F)** BSJs with RBPs binding to the flanking regions, **(G)** BSJs with #$RCS_{across}$ > 0 or RBPs binding to the flanking regions, and
**(H)** minimum distance of RBP-binding sites to BSJs. For (A, B, C, D, E, F, G), the odds ratios represented the ratios of the occurrences of the examined factors of (A, B, C, D, E, F,
G) for non-depleted circRNAs to the occurrences of those for depleted circRNAs. The dashed lines represented odds ratio = 1. Odds ratios and $P$-values were determined
using two-tailed FET. $P$-values were FDR adjusted across 19 mock-treated sample pairs for each examined factor using Benjamini–Hochberg correction. For (H), $P$-values
were determined using WRST (greater or less) and FDR adjusted across 19 mock-treated sample pairs for each examined factor using Benjamini–Hochberg correction. The
Wilcoxon effect sizes and the corresponding 95% confidence intervals are plotted (see also Table S4). The number of mock-treated sample pairs that passed the
statistical significance tests with FDR < 0.05 (or −$\log_{10}$(FDR) > 1.3) are represented in curly brackets.

pairs, which were two replicates generated from the same group but different studies (Table S2) and showed the similar results (Fig 6C, right). Furthermore, regarding the mock samples of the 19 mock-treated sample pairs, we extracted the HeLa-specific circRNAs that were detected in the examined HeLa samples only but not in the non-HeLa cell lines/tissues. For

each of the five HeLa-based mock-treated sample pairs, the HeLa-specific circRNAs were divided into circRNAs detected in single replicate only and circRNAs detected in multiple replicates. Our results revealed that the percentages of not-depleted circRNAs were significantly higher in circRNAs detected in multiple HeLa replicates than in circRNAs detected in single replicate only for all five HeLa-based

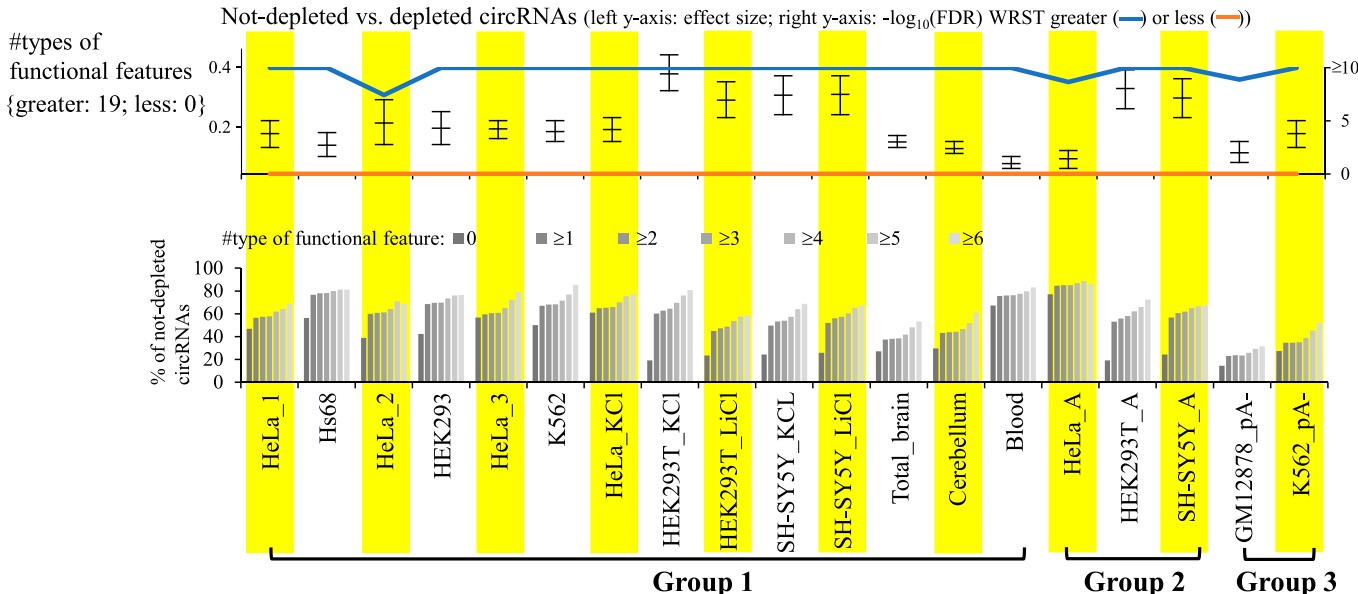

**Figure 5. Impact of functional features on circRNA reliability.**
Nine types of functional features (see the Materials and Methods section) for circRNAs are examined. For the bottom panel, the correlations between percentages of non-depleted circRNAs and number of supporting functional features are shown. *P*-values are determined using WRST (greater or less) and FDR adjusted across 19 mock-treated sample pairs for each examined feature using Benjamini–Hochberg correction. The Wilcoxon effect sizes and the corresponding 95% confidence intervals are plotted (see also Table S4). The number of mock-treated sample pairs that passed the statistical significance tests with FDR < 0.05 (or −$\log_{10}$(FDR) > 1.3) are represented in curly brackets.

pairs (Fig 6D, left). Such a trend was held for the K562-based pairs (Fig 6D, right).

# Discussion

The reliability of computationally detected circRNAs is not guaranteed because of varied types of false positives. It is particularly difficult to pick out spurious BSJ (or circRNA) events derived from in vitro artifacts among a tremendous number of previously identified NCL junctions. Such in vitro artifacts may frequently arise from template switching by random hexamer primed reverse transcriptase (RT) reactions (52), which can switch from one template to another with short homologous sequences at the NCL junction sites (53, 54). Of note, some NCL junctions passing RT-based validations were finally confirmed to be generated from in vitro artifacts by further validations (10). As RNA-seq data are generated from RT-based sequencing strategies, it is unavoidable that a number of spurious NCL events masquerade as genuine BSJs because of such RT-based artifacts (1). We asked whether the approaches based on comparisons of paired mock and treated samples can reflect circRNA reliability to a certain extent. We emphasized that, for the robustness of our analyses, 19 mock-treated sample pairs of different samples/studies based on three types of RNA treatment approaches (Table 1): RNase R approach (group 1), A-tailing RNase R approach (group 2), and non-poly(A) approach (group 3) were performed for the mock-treated comparisons. Because highly structured 3′ ends or G-quadruplexes may affect the effect of the RNase R treatment (32), we also examined

whether the presence of G-quadruplexes across the BSJs affected the percentages of not-depleted circRNAs and found no significant differences between circRNAs with and without removing G-quadruplexes across BSJs (Fig S4 and Table S3). Moreover, we extracted BSJs from two high-confidence circRNA datasets, in which the BSJ events had passed multiple experimental validations. For the first dataset (8), BSJs were supported by both avian myeloblastosis virus- and Moloney murine leukemia virus-derived RNA-seq reads (designated as "RT-independent circRNAs"). RT-independent circRNAs are regarded to be highly accurate because comparisons of different RTase products can effectively detect RT-based artifacts (1, 10, 52, 53). For the second dataset (55), BSJs (or their functions) were previously validated by RT-based experiments and at least one type of non-RT–based experiments simultaneously in at least one human tissues or cell lines (designated as "RT-/non-RT–validated circRNAs"; see the Materials and Methods section). We observed that the percentages of both RT-independent (Fig 7A) and RT-/non-RT–validated (Fig 7B) circRNAs were significantly higher in not-depleted circRNAs than in depleted circRNAs in all mock-treated sample pairs examined. Moreover, we compared circAtlas circRNAs with circRNAs from eight other publicly accessible circRNA databases and identified circAtlas-specific circRNAs. We presumed that circAtlas-specific circRNAs were less probably to be reliable as compared with circRNAs collected in multiple databases. Indeed, the percentages of circAtlas-specific circRNAs were significantly lower in not-depleted circRNAs than in depleted circRNAs (Fig 7C).

For the relative influence of each individual factor on circRNA reliability, our RCVE analysis revealed that "conservation level of circRNA" and "full-length circular sequences" were the top two

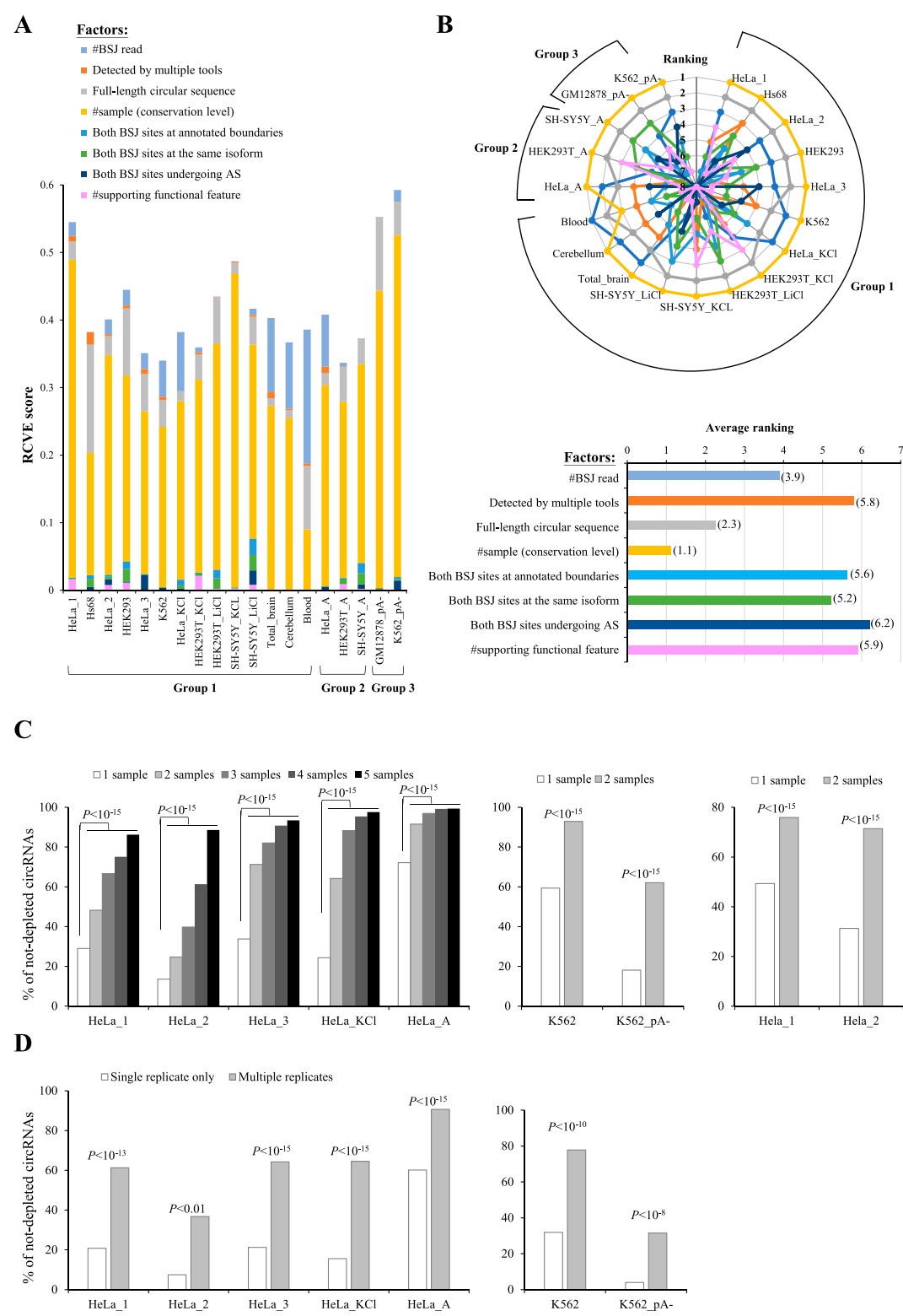

**Figure 6. Assessment of relative influence of each individual factor on circRNA reliability.**
**(A)** RCVE scores of the examined factors for each mock-treated sample pair. The length of each color segment in each bar represents the RCVE score of the corresponding examined factor. **(B)** Ranking (top) and average ranking (bottom; see also the numbers in the parentheses) of the RCVE scores of the examined factors. **(C)** Correlations between the percentages of not-depleted circRNAs and number of samples observed the circRNAs in HeLa-based (left and right) or K562-based (middle) mock-treated sample pairs. Of note, the HeLa_1 and HeLa_2 mock-treated sample pairs were generated from the same group but different studies. **(D)** Comparisons of percentages of not-depleted circRNAs for HeLa-specific circRNA (left) or K562-specific circRNAs (right) detected in single replicate only or multiple replicates. All P-values are determined using two-tailed FET.

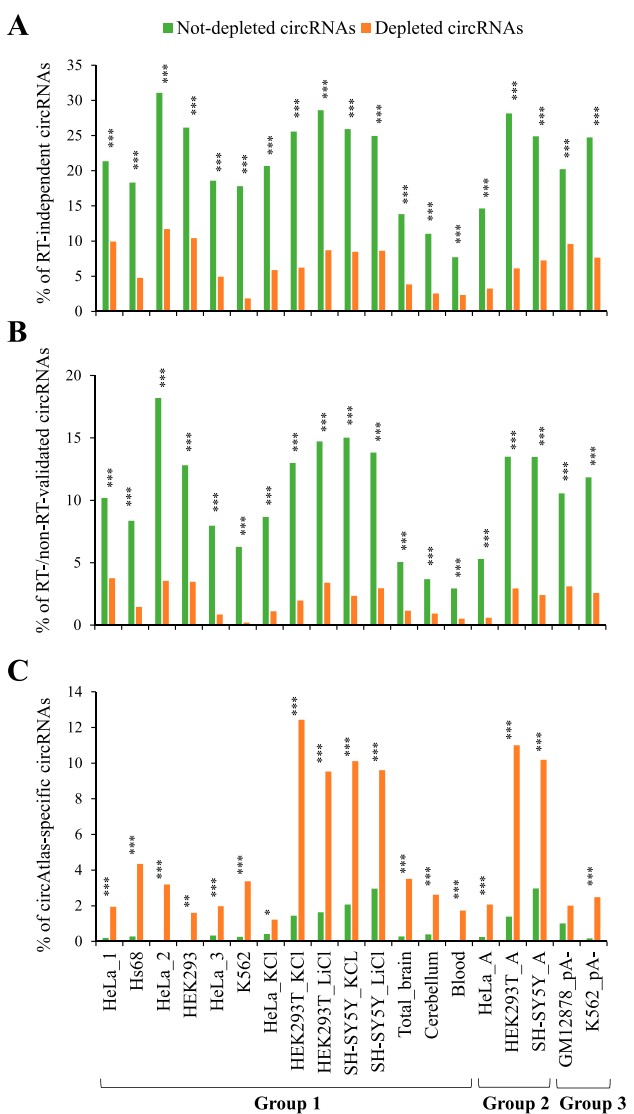

**Figure 7. Robustness analyses of the approaches based on comparisons of paired mock and treated samples for assessing circRNA reliability.**
**(A, B, C)** Comparisons of the percentages of (A) RT-independent circRNAs, (B) RT-/non-RT–validated circRNAs, and (C) circAtlas-specific circRNAs in the not-depleted and depleted circRNAs for all mock-treated sample pairs examined. RT-independent and RT-/non-RT–validated circRNAs represented high-confidence circRNAs (see text and the Materials and Methods section). CircAtlas-specific circRNAs were the circAtlas circRNAs that are not observed in eight other publicly accessible circRNA databases (see the Materials and Methods section). *P*-values were determined using two-tailed FET and FDR adjusted across 19 mock-treated sample pairs for each examined feature using Benjamini–Hochberg correction. *FDR < 0.05. **FDR < 0.01. ***FDR < 0.001.

dominant determinants of circRNA reliability (Fig 6B). For the most important factor in affecting circRNA reliability, because template switching events occur randomly during reverse transcription in vitro, a circRNA candidate observed in multiple samples or replicates is less likely to be generated from such an in vitro artifact. For the second most important factor in affecting circRNA reliability, full-length circular sequences can be reconstructed by Illumina short RNA-seq reads with the reverse overlap (RO) feature (43) or identified by single-molecule long reads (44). With the support of

RO-merged reads, the BSJ events are less probable to originate from template switching because both ends of the paired-end reads are sequencing from the same fragment and reversely overlapped with each other by alignment. Similarly, with the direct support of full-length circular sequences from single-molecule long reads, the nanopore-based full-length circRNAs are less likely to be artifacts derived from template switching.

Our analyses also suggested that "supporting BSJ read count," "supporting functional features," and "BSJs detected by multiple tools" were good indicators for assessing circRNA reliability. Because the circularized sequence is shared with its colinear counterpart except for the BSJ sites, the supporting BSJ read is a unique evidence of charactering a given circRNA event. It is known that most circRNAs are expressed at a low level (3, 4, 5); particularly, a considerable number of detected circRNA candidates were supported by one BSJ read only (e.g., Fig 1E). It is challenging to determine the reliability of such low-abundance circRNAs. Detection tools often set a threshold of supporting BSJ read count to minimize false positives. For example, circRNA_finder (56) and segemehl (57) filter circRNA candidates with a BSJ read count of at least five. In addition to BSJ reads, the functional features of the predicted miRNA/RBP-binding sites or ORFs spanning the BSJs implies the relevance of the BSJ events to functional importance. These features provide another unique support of circRNA reliability. Regarding the factor of "BSJs detected by multiple tools," a previous study reported that combining the results of multiple tools may compensate for the possible weak points of each single tool, and thereby increase the confidence of the detected circRNAs (58). This also reflects our observation of the negative correlation between the percentages of false-positive BSJs derived from alignment ambiguity and the numbers of the detected tools (Fig 1D).

In terms of circRNA biogenesis, we showed that the factors of "both BSJ donor and acceptor splice sites at the same colinear transcript isoforms," "both BSJ donor and acceptor splice sites at the annotated exon boundaries," and "both BSJ donor and acceptor splice sites undergoing annotated AS events" were important for determining circRNA reliability. A possible reason is that circRNAs are also generated by canonical spliceosomal machinery (1, 47, 48, 49). Previous studies also reported that NCL transcript products (including circRNAs, fusion events, and *trans*-spliced transcripts) not matching well-annotated exon boundaries were more likely to be in vitro artifacts (53, 59, 60). Particularly regarding the transcripts generated during the transcriptional process, in addition to *cis*-backsplicing, an observed intragenic NCL junction may also arise from *trans*-splicing (1, 10). Although most of the detected NCL events may be derived from *cis*-backsplicing, previous studies based on comparisons of poly(A)-selected and poly(A)-depleted RNA-seq data suggested that a considerable percentage of detected NCL junctions were derived from *trans*-spliced RNAs (8, 61). Actually, a number of NCL junctions detected in human cells were experimentally validated to be generated by *trans*-splicing rather than *cis*-backsplicing (10, 53, 62, 63). As *trans*-splicing occurs between two or more separate precursor mRNAs (64, 65), it is possible that an observed NCL junction with its two splice sites from different colinear transcript isoforms is derived from a *trans*-splicing event. We observed that the full-length circRNAs with the support of RO-merged short reads or single-molecule long reads, which were

principally not generated from *trans*-splicing, were significantly enriched for the BSJs with donor and acceptor splice sites at the same colinear transcript isoforms (Fig S5).

On the other hand, we showed that the factors related to RCS and RBPs binding to the flanking regions of BSJs were not good indicators for assessing circRNA reliability. Our observations reflect a previous notion that the existence of RCS is neither sufficient nor necessary for *cis*-backsplicing (2). Although some circRNA cases were demonstrated to be facilitated by base-paring between RCSs at two flanking intronic sequences of the BSJs (3, 8, 29), only a limited number of RCS-circRNA correlations were experimentally confirmed. Multiple RNA parings derived from different RCS pairs across flanking introns or within individual flanking introns can compete against each other and lead to the reduction of circRNA production (29, 47, 66) or alternative *cis*-backsplicing (30, 67), further complicating the determination and validation of the RCS-circRNA correlations. In addition, although repetitive elements abundantly emerge in mammalian genomes, which contribute to RNA pairing, circRNAs observed in non-mammalian species (e.g., *Drosophila melanogaster* (56) and *Oryza sativa* (68)) are often lacking the feature of RCS. For the associations between RBPs and circRNA formation, different RBPs may play opposite roles (up- or down-regulation) in regulating circRNA expression. Some RBPs, such as FUS (69) and ADARs (70), were even showed to mediate circRNA production in a bidirectional manner. Because different RBPs may have overlapping capacities for binding to RCSs, it requires further investigations to understand the joint (coordinate or competitive) effects of different RBPs on the regulations for each individual circRNA (2, 71).

This study systematically assessed the impacts of a dozen factors related to identification, conservation, biogenesis, and function on circRNA reliability on the basis of comparisons of mock-treated sample pairs from three different RNA treatment approaches. Of the examined factors, eight were shown to be the important indicators of circRNA reliability in all mock-treated sample pairs examined, regardless of the RNA treatment approaches. We assessed the relative influence of each individual factor on circRNA reliability and suggested that the most important factors in descending order were the conservation level of circRNA, full-length circular sequences, supporting BSJ read count, both BSJ donor and acceptor splice sites at the same colinear transcript isoforms, supporting functional features, both BSJ donor and acceptor splice sites at the annotated exon boundaries, BSJs detected by multiple tools, and both BSJ donor and acceptor splice sites undergoing annotated AS events. All the circAtlas circRNA candidates and the corresponding factors examined were provided (Table S3). Importantly, we found a remarkably positive correlation between number of supporting factors and the percentage of high-confidence circRNAs (i.e., RT-independent and RT-/non-RT–validated circRNAs) and a remarkably negative correlation between number of supporting factors and the percentage of circAtlas-specific circRNAs (Fig 8). This revealed the additive effects of these factors on circRNA reliability. Particularly, more than 70% of circRNAs without any supporting factors were circAtlas-specific circRNAs, whereas such a percentage sharply declined to ≤3% when considering the circRNAs with at least five supporting factors (Fig 8), supporting the effectiveness of these factors in determining circRNA reliability before performing experimental validations. To

the best of our understanding, this study, for the first time, presents a useful guideline to systematically assess circRNA reliability with simultaneously accounting for varied types of factors, facilitating further functional investigation of this important class of non-canonical transcripts.

# Materials and Methods

## CircRNA candidates and RNA-seq data collection

As shown in Fig 1A, human circRNA candidates examined (Table S1) were extracted from circAtlas 2.0 (22), which contained 580,654 human circRNA candidates based on 240 samples collected from 19 tissues across seven vertebrate species. The related information such as circRNA identification factors (e.g., circRNA detection tools and full-length circular sequences) and conservation factors (e.g., conservation of circRNAs across species, tissues, and samples) were also downloaded from circAtlas 2.0. CircAtlas circRNAs were identified by CIRI2 (38, 72), find_circ (28), CIRCexplorer (73), or DCC (74). The full-length circular sequences were reconstructed by CIRI-full (43). All the analyses were based on the human reference genome (GRCh38) and the Ensembl annotation (version 100). The circRNA candidates that were potentially derived from alignment ambiguity (circRNA candidates had an alternative colinear explanation or multiple matches against the human genome) were detected by NCLcomparator (35) with default parameters and were not included in the subsequent analyses for accuracy (Fig 1A). After that, 480,471 circAtlas circRNAs were retained. The remaining RNA-seq reads were aligned against the human reference genome using STAR with default parameters (75). For each circRNA (BSJ), we generated a pseudo sequence by concatenating the sequences flanking the BSJ (within −50 nucleotides of the donor site to +50 nucleotides of the acceptor site) and aligned all STAR-unmapped reads against the 100-bp BSJ-based pseudo sequence using BWA with default parameters (76). A STAR-unmapped read was determined as a BSJ read if the read matched to ≥80% of the BSJ-based pseudo sequence and spanned the junction boundary by ≥10 bp on both sides of the BSJ.

The mock-treated sample pairs (19 pairs) were extracted from publicly accessible databases and categorized into three groups according to the RNA treatment approaches (see Tables 1 and S2):

Group 1: RNase R approach, mock sample versus RNase R–treated sample (14 pairs).
Group 2: A-tailing RNase R approach, mock sample versus A-tailing RNase R–treated (RNAs treated with coupling A-tailing and RNase R digestion) sample (three pairs).
Group 3: non-poly(A) approach, mock sample versus non-poly(A)–selected (RNAs treated with depletion of both ribosomal RNAs and polyadenylated RNAs) sample (two pairs).

Of note, the downloaded mock-treated sample pairs should simultaneously satisfy the following criteria: (1) the number of the circRNA candidates detected in the mock samples should be greater than 600 and (2) more than one-third of the circRNA candidates detected in the mock samples should be detected in the

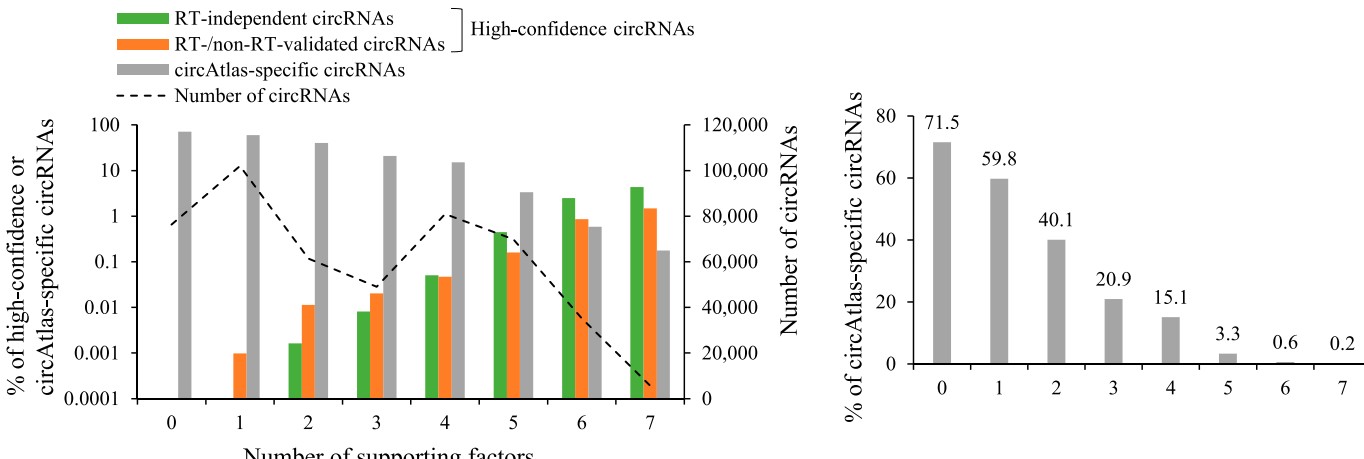

**Figure 8. The correlation between number of supporting factors and the percentage of high-confidence circRNAs (RT-independent or RT-/non-RT–validated circRNAs) and circAtlas-specific circRNAs for the 480,471 circAtlas circRNA candidates.**
The right panel showed a clearer graph of the correlation between number of supporting factors and the percentage of circAtlas-specific circRNAs. The eight important factors illustrated in Fig 6 except for the factor of "supporting BSJ read count" were considered because this factor was dependent on the examined samples and the corresponding RNA-seq data. For the factors of "number of samples" and "number of supporting functional features," three was used as a cutoff value. The detailed information can be found in Table S3.

corresponding treated samples. For each RNA-seq dataset, duplicated RNA-seq reads were removed using FastUniq (77).

## Treat/mock ratio calculation

For accuracy, only the circRNAs detected in the mock samples with the support of at least two BSJ reads were considered in the analysis for each mock-treated sample pair. The expression levels of circRNAs were calculated using the BSJ reads per million raw reads (RPM). For each considered circRNA candidate, we calculated the ratio of the circRNA expression detected in the treated samples to that detected in the corresponding mock samples (i.e., treat/mock ratio). The not-depleted and depleted circRNAs are defined as follows:

$$\text{treat/mock ratio} \begin{cases} \geq 1, \text{ Not} - \text{depleted circRNAs} \\ < 1, \text{ Depleted circRNAs.} \end{cases}$$

## Extractions of various factors

The factors of CIRI-full–identified full-length circular sequence, number of circRNA detection tools, conservation patterns at the individual (or sample), tissue, and species levels (including tissue specificity index) were extracted from circAtlas 2.0 (22). The BSJ events with the direct support of full-length circular sequences from nanopore long RNA-seq reads were downloaded from Liu et al's study (44). The evolutionary rates were determined by phyloP (45) or phastCons (46) scores, which were downloaded from the UCSC Genome Browser at https://genome.ucsc.edu/. For each BSJ, we considered the four regions around the BSJ: (1) within +1 to +10 nucleotides of the acceptor site; (2) within −10 to −1 nucleotides of the acceptor site; (3) within −10 to −1 nucleotides of the donor site;

and (4) within +1 to +10 nucleotides of the donor site. The evolutionary rate of each region was measured by the average value of the phyloP (or phastCons) scores of the considered nucleotides (10 bp) within the region. The annotated exon boundaries and alternative splicing (AS) patterns were derived from the Ensembl annotation (version 100). The splice site strength of BSJs was estimated using MaxEntScan based on three scoring models (maximum entropy model, first-order Markov model, and weight matrix model) (78). The numbers of reverse complementary sequences (RCSs) across the flanking sequences ($RCS_{across}$; ±20,000 nucleotides of the BSJs) or within individual flanking sequences ($RCS_{within}$) of the BSJs were calculated using CircMiMi (79) with the command line: circmimi_tools check RCS –dist 20,000 genome.fa circRNAs.tsv out.tsv. RBPs binding to the flanking regions (±1,000 nucleotides) of BSJs was determined according to CLIP-supported RBP-binding sites downloaded from ENCORI (80) at http://starbase.sysu.edu.cn/. G-quadruplexes across the BSJs were determined using the QGRS mapping algorithm with default parameters (81, 82). The predicted G-quadruplexes should span the BSJ boundaries by ≥5 bp on both sides of the BSJs. miRNA-binding sites across the BSJs were determined using miRanda 3.3a (83) with pairing score ≥ 155. RBP-binding sites across the BSJs were determined using RBPmap (84) with a high stringency level and conservation filter. The predicted miRNA (or RBP)-binding sites should span the BSJ boundaries by ≥5 (or ≥2) bp on both sides of the BSJs, respectively. The seven types of evidence for coding potential of more than 320,000 human circRNAs (including circAtlas circRNAs) were extracted from TransCirc (85). The seven types of evidence included ribosome/polysome-binding evidence, experimentally supported translation initiation site, internal ribosome entry site, predicted m6A modification site, circRNA-specific ORF, sequence composition score for the ORF, and mass spectrometry data–supported peptide across the BSJ. The translatable scores of circRNAs were downloaded from TransCirc.

### Extractions of high-confidence circRNAs and circAtlas-specific circRNAs

The high-confidence circRNAs examined here included RT-independent and RT-/non-RT–validated circRNAs (see text). The RT-independent circRNAs (1,478 events), which were supported by both avian myeloblastosis virus- and Moloney murine leukemia virus-derived RNA-seq reads, were extracted from our previous study (8). The RT-/non-RT–validated circRNAs (553 events), in which the NCL junctions (or their functions) were validated by RT-based experiments and at least one type of non-RT–based experiments in human tissues/cell lines, were extracted from CircR2disease (version 2.0) (55). The circAtlas-specific circRNA candidates were the BSJ events that were collected in circAtlas only rather than in eight other publicly accessible circRNA databases (including CIRCpedia v2 (86), CircRic (87), MiOncoCirc v2.0 (88), TSCD (51), circBase (89), circRNADb (90), and exoRBase 2.0 (91)). All the circRNA candidates deposited in CircR2disease or non-circAtlas circRNA databases were not considered if the NCL junction coordinates were not available or cannot be converted to the genomic coordinates on the GRCh38 assembly by liftOver (92). For the NCL junctions with non-assigned strands, the junction coordinates were matched to the circRNA candidates in circAtlas. The analyzed circRNA candidates, the related factors, and the treat/mock ratios of the corresponding mock-treated sample pairs are given in Table S3. All the coordinates in the table were one-based.

### Statistics analysis

For enrichment analyses, we compared not-depleted circRNAs with depleted ones in terms of various factors related to circRNA identification, conservation, biogenesis, and function. $P$-values were determined using two-tailed Fisher's exact test or Wilcoxon rank-sum test (WRST; greater and less) (see also the figure legends). $P$-values were further FDR adjusted across 19 mock-treated sample pairs for each examined feature using Benjamini–Hochberg correction. The Wilcoxon effect sizes and the corresponding 95% confidence intervals were calculated with the wilcox_effsize function implemented in the rstatix R package (93). To measure the relative effect of individual factor in determining circRNA reliability, we first employed a GLM with all the examined factors using the statsmodels package (0.13.2) (94) in Python and then performed the RCVE (50) as follows:

$$\text{glm}\,(y \sim f_1 + f_2 + f_3 + f_4 + f_5 + f_6 + f_7 + f_8, \text{family = binomial})$$

$$RCVE = \frac{r_{\text{all}}^2 - r_{\text{reduced}}^2}{r_{\text{all}}^2}$$

In the GLM, $y$ indicated whether a circRNA candidate was a not-depleted circRNA (i.e., not-depleted circRNA = 1; depleted circRNA = 0). $f_1 \sim f_8$ represented the eight factors examined in the mode: supporting BSJ read count; whether BSJs were detected by multiple tools; whether BSJs can be reconstructed full-length circRNAs; number of samples observed the circRNAs; whether BSJs agreed to annotated exon boundaries; whether BSJs had both donor and acceptor splice sites matching annotated exon boundaries from the same colinear transcript isoforms; whether BSJs were subject to AS; and number of supporting functional features, respectively. In the RCVE formula, $r_{\text{all}}^2$ and $r_{\text{reduced}}^2$ represented the $r^2$ values calculated by the GLM including all the eight factors and excluding the factor of interest, respectively. The RCVE scores range from 0 to 1, with a higher RCVE score indicating a more important contribution of the examined factor to the model. The detailed results of the statistical significance tests for each examined factor are given in Table S4.

## Data Availability

The data underlying this study are available in the text and in Tables S1–S4. Tables S1–S4 and all the codes used to generate the results are available at GitHub (https://github.com/TreesLab/circRNA_features).

## Supplementary Information

## Acknowledgements

This work was supported by T-J Chuang, Genomics Research Center, Academia Sinica, Taiwan and the National Science and Technology Council, Taiwan (MOST 111-2311-B-001-021). We thank the related organizations for generating and providing the RNA-seq data used in this study. We also thank other members of T-J Chuang laboratory for helpful discussions.

### Author Contributions

T-J Chuang: conceptualization, formal analysis, investigation, methodology, and writing—original draft, review, and editing.
T-W Chiang and C-Y Chen: software and methodology.

### Conflict of Interest Statement

The authors declare that they have no conflict of interest.

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
