## [Reviewer comments · Life Science Alliance]

Life Science Alliance

Assessing the impacts of various factors on circular RNA reliability

Trees-Juen Chuang, Tai-We Chiang, and Chia-Ying Chen

DOI: <https://doi.org/10.26508/lsa.202201793>

Corresponding author(s): Trees-Juen Chuang, Genomics Research Center, Academia Sinica

Review Timeline:

Submission Date:	2022-11-01
Editorial Decision:	2022-12-01
Revision Received:	2023-01-31
Editorial Decision:	2023-02-14
Revision Received:	2023-02-15
Accepted:	2023-02-15

Scientific Editor: Novella Guidi

Transaction Report:

December 1, 2022

Re: Life Science Alliance manuscript #LSA-2022-01793-T

Assistant Research Fellow Trees-Juen Chuang
Genomics Research Center
Genomics Research Center, Academia Sinica, Taipei 11529, Taiwan
Taipei 11529
Taiwan

Dear Dr. Chuang,

Thank you for submitting your manuscript entitled "Assessing the impacts of various factors related to identification, conservation, biogenesis, and function on circular RNA reliability" to Life Science Alliance. The manuscript was assessed by expert reviewers, whose comments are appended to this letter. We invite you to submit a revised manuscript addressing the Reviewer comments.

Thank you for this interesting contribution to Life Science Alliance. We are looking forward to receiving your revised manuscript.

Sincerely,

B. MANUSCRIPT ORGANIZATION AND FORMATTING:

Reviewer #1 (Comments to the Authors (Required)):

Comments to authors

In the manuscript entitled 'Assessing the impacts of various factors related to identification, conservation, biogenesis, and function on circular RNA reliability' (LSA-2022-01793-T), the authors performed a series of thorough analyses for a comprehensive set of circRNA candidates to fill the gap between experimental results and bioinformatic predictions. The assessments were performed by using multiple data sources and independent tools. The results of this study will be a great addition and asset to the field of circRNA or related fields studying non-coding RNA if the authors can further clarify some ambiguous parts or statements listed in the following section.

Major points

1. The description regarding the results of the consistence among identified circRNA candidates by four tools is not accurate -- "the majority (63.5%) of the examined circRNA candidates were detected by only one or two tools (Fig. 1B)." The authors should also consider in an alternative aspect, 63.5% (27+4.6+30.9) of the candidates were identified by at least two tools. Even just counting results from 3 and 4 tools, 35.5% is somehow not far from 36.5% (1 tool). A softer statement will be preferred.
2. To better elaborate the case(s) of false positive candidates arising from alternative co-linear scenarios in the Figure 1C, the authors should show a few examples providing both sequence alignment and illustration, perhaps in supplementary data. This will definitely convince the readers that such discrepancies are indeed problematic and should be carefully evaluated.
3. The percentages of "with an alternative co-linear explanation" show in Figure 1D should be shown on the top of each bar.
4. Related to Figure 1D and the last line in page 8 "For accuracy, we excluded the 100,183 circRNAs and considered the remaining circRNAs (480,471 events) for the following analyses." --- are these two groups referred to " 'not-depleted' and 'depleted' circRNAs" in Figure 1A and the rest of Figures. If yes, the authors should declare it here.
5. It has too much information in the Figure 1E containing both % and numbers. The plot should be split to two, one for numbers (all, larger or equal to 2, =1) and the other for the percentages (larger or equal to 2, =1).
6. It might not be easy for readers to catch up what these 19 groups of bars listed in Figure 2. It will helps a lot to describe in the figure legend 2A at very beginning, or add additional labels/lines/frames in each plot. Similar problems were found for Figure 3, 4 and 5.
7. The way for estimation of FDR in the Figure 2A and 3 should be provided.
8. Although the U2 or U12 preference has been reported previously, the conclusion of no preference was based on early datasets perhaps biased by some false positive candidates. The authors have taken the splicing signal into consideration for reliability of circRNA candidates (Figure 4A and 4B). The authors may easily re-evaluate this previous conclusion using their datasets with more reliability. This will help a lot to clarify this issue for the circRNA community.
9. On page 31, the authors stated "we showed that the factors related to RCS and RBPs binding to the flanking regions of BSJs were not good indicators for assessing circRNA reliability." - the reviewer simply agreed with this observation. However, it will be more informative if the authors provided details regarding what kind of RBPs were involved in the analyses in the Figure 4F or what kind of RBPs in terms promoting or suppressing splicing. Is there any particular metrics for the distances to backsplicing sites? Are these RBP sites predicted or based on experimental results such as PAR-CLIP?
10. Related to point 9, the poor assessment for circRNA reliability by RCS or RBP was caused by taking each alone for assessment. Because the biogenesis of circRNA can be promoted by either RBP or RCS, flanking regions without RBP sites may have RCS for circRNA biogenesis. Thus it may be more relevant to have 'RCS/RBP'+ vs 'RCS/RBP'- groups for assessment instead of RCS+ vs RCS- or RBP+ vs RBP-.
11. The authors described that highly structured RNA or G-quadruplexes may compromise the enzymatic reaction and thus lead to some false positive candidates. There are a few bioinformatic tools for predicting G-quadruplexes such as allquads, QGRS Mapper, G4Hunter, G4Boost and Quadparser. It will be recommended that the authors include at least one of these analyses along with Figure 4 if it is feasible.
12. The meaning of color code in the Figure 6A is ambiguous. What is the implication for the length of each color segment in each bar to RCVE score?
13. For the figure 6C and D, What are the p-values calculated from? There were two types of measurements mentioned - numbers and correlation.

Minor points

>> a few typing errors:

..circRNA candidates were identified by four bioinformatics tools including.. (page 7, last line) -> 'bioinformatic tools'

Referee Cross-Comments:

The reviewer 2's comments are constructive and helpful to this study.

Reviewer #2 (Comments to the Authors (Required)):

In this study, the authors performed a comprehensive investigation of factors on circRNA identification reliability. Indeed, the authors concluded the importance of these factors through a generalized linear model. Briefly, the contents of the manuscript fulfills the title adequately. Here are some comments listed below:

- 1.The first section of results in the paper is more like an overview of the data and conditions to select the analyzed circRNA. I suggest the authors move this section to Materials and methods.
- 2.Even though the statistic test are significant between non-depleted and depleted circRNAs, the differences are small. I suggest the authors to use effect size to confirm the tests' reliability.
- 3.The authors performed a GLM analysis for the selected 8 factors to further compare their importance. I am curious about the results on performing a GLM analysis for all the factors to check if these eight factors are still significant to the circRNA identification reliability.
- 4.If the authors can make more efforts to investigate the reason why the identified factors are significant to the circRNA identification reliability, the scientific advance of this study would be largely improved.

We deeply thank both reviewers for their constructive comments and suggestions. We present our responses to each individual comment below. The modifications are also highlighted with red color in the text.

Reviewer 1:

In the manuscript entitled 'Assessing the impacts of various factors related to identification, conservation, biogenesis, and function on circular RNA reliability' (LSA-2022-01793-T), the authors performed a series of thorough analyses for a comprehensive set of circRNA candidates to fill the gap between experimental results and bioinformatic predictions. The assessments were performed by using multiple data sources and independent tools. The results of this study will be a great addition and asset to the field of circRNA or related fields studying non-coding RNA if the authors can further clarify some ambiguous parts or statements listed in the following section.

1. The description regarding the results of the consistence among identified circRNA candidates by four tools is not accurate -- "the majority (63.5%) of the examined circRNA candidates were detected by only one or two tools (Fig. 1B)." The authors should also consider in an alternative aspect, 63.5% (27+4.6+30.9) of the candidates were identified by at least two tools. Even just counting results from 3 and 4 tools, 35.5% is somehow not far from 36.5% (1 tool). A softer statement will be preferred.

Answer:

We thank the reviewer for the valuable suggestion. Accordingly, the sentence "the majority (63.5%) of the examined circRNA candidates were detected by only one or two tools (Fig. 1B)" was modified as "a considerable percentage (36.5%) of the examined circRNA candidates were detected by only one tool (Fig. 1B)". (Lines 28-29, Page 4)

2. To better elaborate the case(s) of false positive candidates arising from alternative co-linear scenarios in the Figure 1C, the authors should show a few examples providing both sequence alignment and illustration, perhaps in supplementary data. This will definitely convince the readers that such discrepancies are indeed problematic and should be carefully evaluated.

Answer:

We thank the reviewer for the comment. Accordingly, a new supplementary figure (Supplementary Fig. 1) was provided, which presented three BSJ cases that have an alternative co-linear explanation.

BSJ-1: chr1:27732657+:chr1:27730232:+ (host gene: FAM76A)

BLAT-alignment result with an alternative co-linear explanation:

```

00000001 ttcagatattctaagaaaacctgtgtctctttctcctaagtaaaa 00000050
>>>>>> |||||||||||||>>>>>>>>>>>>
27732558 ttcagatattctaagaaaacctgtgtctctttcctaagtaaaa 27732607
                                                    } Exon4 of FAM76A

00000051 ccaatacaatatgcaagaaatgtgctcagaacgtgcagttgatggaacg 00000100
>>>>>>> |||||||||||||>>>>>>>>>>>>
27732608 ccaatacaatatgcaagaaatgtgctcagaacgtgcagttgatggaacg 27732657

00000101 gctagagtgaatgacgatctcggctcactgcaacctccactcccag 00000150
>>>>>>> |||||||||||||>>>>>>>>>>>>
27736995 gctagagtgaatgacgatctcggctcactgcaacctccactcccag 27737044
                                                    } Intron6 of FAM76A

00000151 gttcaagcattctcctgcctcagcctcccgagtagctggattacagc 00000200
>>>>>>> |||||||||||||>>>>>>>>>>>>
27737045 gttcaagcattctcctgcctcagcctcccgagtagctggattacagc 27737094

```

BSJ-2: chr7:65388901+:chr7:65387092:+ (host gene: ZNF92)

BLAT-alignment result with an alternative co-linear explanation:

```

00000001 acaggtattgtcttctaagccagacctgatactgctggagcaagg 00000050
>>>>>> |||||||||||||>>>>>>>>>>>>
64918193 acaggtattgtcttctaagccagacctgatactgctggagcaagg 64918242
                                                    } Exon3 of ZNF273

00000051 aaaagagccctggaatctgaagagacatgagatgtagcaaaaccccag 00000100
>>>>>>> |||||||||||||>>>>>>>>>>>>
64918243 aaaagagccctggaatctgaagagacatgagatgtagcaaaaccccag 64918292

00000101 ggaccactgacatttaggatgtgaaatagaattctcttagaggaatg 00000150
>>>>>>> |||||||||||||>>>>>>>>>>>>
65387902 ggaccactgacatttaggatgtgaaatagaattctcttagaggaatg 65387951
                                                    } Exon2 of ZNF92

00000151 gcaatgcctggaactgcaacggaatttatagagatgtagttag 00000200
>>>>>>> |||||||||||||>>>>>>>>>>>>
65387952 gcaatgcctggaactgcaacggaatttatagagatgtagttag 65388001

```

BLAT-alignment result with an alternative co-linear explanation:

```

0000001 atttatggctaagaagttttcactggatgcgttaataaccatgttta 0000050
<<<<<<< ||||||||||||||||||||||||||||||||||||||| <<<<<<<<
99310864 atttatggctaagaagttttcactggatgcgttaataaccatgttta 99310815
} Exon8 of ADH1B

0000051 cctttgaaaaataaatgaaggatttgacctgttcactctgggaaaag 0000100
<<<<<<< ||||||||||||||||||||||||||||||||||||||| <<<<<<<
99310814 cctttgaaaaataaatgaaggatttgacctgttcactctgggaaaag 99310765
} Exon2 of ADH1A

0000101 gtaatcaaatgcaaacgagctgtgctatggaggtaaagaaccttttc 0000150
<<<<<<< ||||||||||||||||||||||||||||||||||||||| <<<<<<<
99287665 gtaatcaaatgcaaacgagctgtgctatggaggtaaagaaccttttc 99287616

0000151 cattgaggatgtggagttgcacctcctaaggcttatgaagttcgatta 0000200
<<<<<<< ||||||||||||||||||||||||||||||||||||||| <<<<<<<
99287615 cattgaggatgtggagttgcacctcctaaggcttatgaagttcgatta 99287566

```

Supplemental Figure S1. Examples of BSJ candidates (BSJ-1, BSJ-2, and BSJ-3) with an alternative co-linear explanation. For BSJ-1 (Exon4-Exon3, *FAM76A*), the concatenated sequence has an alternative co-linear explanation (Exon4-Intron6, *FAM76A*). For BSJ-2 (Exon3-Exon2, *ZNF92*), the concatenated sequence has an alternative co-linear explanation (Exon3 (*ZNF273*)-Exon3 (*ZNF92*)). For BSJ-3 (Exon8-Exon2, *ADH1B*), the concatenated sequence has an alternative co-linear explanation (Exon2 (*ADH1B*)-Exon2 (*ADH1A*)).

3. The percentages of "with an alternative co-linear explanation" show in Figure 1D should be shown on the top of each bar.

Answer:

We thank the reviewer for the comment. The percentages of "with an alternative co-linear explanation" and "with multiple hits" were shown for each bar, respectively. The Figure 1D was modified as follows.

Figure 1D. Comparisons of the percentages of circRNA candidates derived from potential alignment ambiguity with an alternative co-linear explanation or multiple hits for the candidates detected by 1, 2, 3, or 4 tools.

4. Related to Figure 1D and the last line in page 8 "For accuracy, we excluded the 100,183 circRNAs and considered the remaining circRNAs (480,471 events) for the following analyses." --- are these two groups referred to " 'not-depleted' and 'depleted' circRNAs" in Figure 1A and the rest of Figures. If yes, the authors should declare it here.

Answer:

We apologize for the unclear description. No, these two groups (100,183 vs. 480,471 events) are not referred to " 'not-depleted' and 'depleted' circRNAs" in Figure 1A and the rest of Figures. Of the 580,654 extracted circRNA candidates, we removed the circRNA candidates that arose from potential alignment ambiguity (100,183 events) and only considered the remaining circRNAs (480,471 events) for the following analyses of circRNA reliability. The sentence was modified as follows.

“For accuracy, we excluded the circRNA candidates arising from potential alignment ambiguity (100,183 events) and considered the remaining circRNAs (480,471 events) for the following analyses of circRNA reliability.” (Lines 1-4, Page 5)

The not-depleted and depleted circRNAs were defined as follows:

$$\text{treat/mock ratio} \begin{cases} \geq 1, \text{ Not-depleted circRNAs} \\ < 1, \text{ Depleted circRNAs} \end{cases}$$

Here the treat/mock ratio is the ratio of the circRNA expression detected in the treated samples to that detected in the corresponding mock samples. (Lines 2-9, Page 14; Materials and Methods)

Figure 1A was also modified accordingly.

Figure 1A. Flowchart of the overall analyses.

- It has too much information in the Figure 1E containing both % and numbers. The plot should be split to two, one for numbers (all, larger or equal to 2, =1) and the other for the percentages (larger or equal to 2, =1).

Answer:

We thank the reviewer for the comment. For simplicity, Figure 1E was modified as follows.

Figure 1E. Comparisons of normalized numbers of circRNA candidates with supporting BSJ read count =1 or ≥ 2 in all extracted mock samples of the 19 mock-treated sample pairs. For each sample, the percentage of circRNA candidates supported by one BSJ read was shown.

We can find that a considerable percentage of circRNAs (38%~80%) were supported

by only one BSJ read

6. It might not be easy for readers to catch up what these 19 groups of bars listed in Figure 2. It will help a lot to describe in the figure legend 2A at very beginning, or add additional labels/lines/frames in each plot. Similar problems were found for Figure 3, 4 and 5.

Answer:

We thank the reviewer for the comment. We have added some sentences in the figure legends. For Figs. 2-5, background colors were added to highlight the examined samples.

7. The way for estimation of FDR in the Figure 2A and 3 should be provided.

Answer:

We thank the reviewer for the comment. All *P* values were FDR adjusted across 19 mock-treated sample pairs for each examined feature using Benjamini-Hochberg (BH) correction in Figures 2-5. The statement was added in the figure legends of these figures.

8. Although the U2 or U12 preference has been reported previously, the conclusion of no preference was based on early datasets perhaps biased by some false positive candidates. The authors have taken the splicing signal into consideration for reliability of circRNA candidates (Figure 4A and 4B). The authors may easily re-evaluate this previous conclusion using their datasets with more reliability. This will help a lot to clarify this issue for the circRNA community.

Answer:

We thank the reviewer for the comment. Since circRNA candidates without the canonical splicing signals (e.g., the U2 “GT-AG” splicing signal) were suggested to be derived from sequencing artifacts [1], most circRNA detectors, including the four tools (CIRI2, find_circ, circExplorer, and DCC) that were used to identify the CircAtlas circRNA candidates examined in this study, utilized the canonical splicing signals at BSJs as a filter. In other words, all candidates with non-canonical splicing signals were not included in CircAtlas. Therefore, the factor of the U2/U12 splicing signals at BSJs is not appropriate to evaluate circRNA reliability of the circRNA candidates.

9. On page 31, the authors stated "we showed that the factors related to RCS and RBPs binding to the flanking regions of BSJs were not good indicators for assessing circRNA reliability." - the reviewer simply agreed with this observation. However,

it will be more informative if the authors provided details regarding what kind of RBPs were involved in the analyses in the Figure 4F or what kind of RBPs in terms promoting or suppressing splicing. Is there any particular metrics for the distances to backsplicing sites? Are these RBP sites predicted or based on experimental results such as PAR-CLIP?

Answer:

We thank the reviewer for the comment. Yes, the RBP sites were based on PAR-CLIP data. RBPs binding to the flanking regions ($\pm 1,000$ nucleotides) of BSJs was determined according to CLIP-supported RBP binding sites downloaded from ENCORI [2] at <http://starbase.sysu.edu.cn/>. The RBPs involved in this study were newly added in Supplemental Table S2. We also considered the minimum distance of the detected RBP binding sites to backsplicing sites and observed that the distance was only slightly shorter in not-depleted circRNAs than in depleted ones (see the newly added Fig. 4H).

Figures 4G and 4H. Impact of factors related to circRNA biogenesis on circRNA reliability: **(G)** BSJs with $\#RCS_{across} > 0$ or RBPs binding to the flanking regions and **(H)** minimum distance of RBP binding sites to BSJs. For (G), the odds ratios represented the ratios of the occurrence of the examined factor for non-depleted circRNAs to the occurrence of that for depleted circRNAs. The dashed lines represented odds ratio=1. Odds ratios and P values were determined using two-tailed Fisher’s exact test. P values were FDR adjusted across 19 mock-treated sample pairs for each examined factor using Benjamini-Hochberg correction. For (H), P values were determined using Wilcoxon rank-sum test (greater or less) and FDR adjusted across 19 mock-treated sample pairs for each examined factor using Benjamini-Hochberg correction. The Wilcoxon effect sizes and the corresponding 95% confidence intervals were plotted (see also Supplemental Table S4). The number of mock-treated sample pairs that passed the

statistical significance tests with $FDR < 0.05$ (or $-\log_{10}(FDR) > 1.3$) were represented in curly brackets.

10. Related to point 9, the poor assessment for circRNA reliability by RCS or RBP was caused by taking each alone for assessment. Because the biogenesis of circRNA can be promoted by either RBP or RCS, flanking regions without RBP sites may have RCS for circRNA biogenesis. Thus it may be more relevant to have 'RCS/RBP'+ vs 'RCS/RBP'- groups for assessment instead of RCS+ vs RCS- or RBP+ vs RBP-.

Answer:

We thank the reviewer for the comment. Accordingly, we examined the BSJs with RCS_{across} or RBP binding sites in flanking regions and did not observe a significant enrichment for not-depleted circRNAs (see the newly added Fig. 4G), also supporting that the features of RCS or RBPs binding to the flanking regions were not good indicators for evaluating circRNA reliability.

11. The authors described that highly structured RNA or G-quadruplexes may compromise the enzymatic reaction and thus lead to some false positive candidates. There are a few bioinformatic tools for predicting G-quadruplexes such as allquads, QGRS Mapper, G4Hunter, G4Boost and Quadparser. It will be recommended that the authors include at least one of these analyses along with Figure 4 if it is feasible.

Answer:

We thank the reviewer for the comment. We emphasized that, for the robustness of our analyses, 19 mock-treated sample pairs of different samples/studies based on three types of RNA treatment approaches (Table 1): RNase R approach (Group 1), A-tailing RNase R approach (Group 2) and non-Poly(A) approach (Group 3) were performed for the mock-treated comparisons. Of note, since highly structured 3' ends or G-quadruplexes may be resistant to RNase R and thereby affect the effect of the RNase R treatment [3], the A-tailing RNase R approach (i.e., Group 2, which was demonstrated to efficiently digest most the RNase R-resistant co-linear RNAs [3]) was considered in our mock-treated comparisons. According to the reviewer's suggestion, we also examined whether the presence of G-quadruplexes across the BSJs affected the percentages of not-depleted circRNAs and found no significant differences between circRNAs with and without removing G-quadruplexes across BSJs (Supplemental Fig. S4 and Supplemental Table S3; see Lines 27-35, Page 9). Here we used the QGRS mapping algorithm (a stand-alone version of QGRS Mapper) with default parameters [4, 5] to examine the presence of G-quadruplexes across the BSJs. The predicted G-quadruplexes should span the BSJ boundaries by ≥ 5 bp on both sides of the BSJs

(Lines 32-34, Page 14; Materials and Methods).

Supplemental Figure S4. Percentages of not-depleted circRNAs with/without removing G-quadruplexes across BSJs. No significant differences between circRNAs with and without removing G-quadruplexes across BSJs were observed (all P values > 0.05). P values were determined using two-tailed Fisher’s exact test. ns, not significant.

12. The meaning of color code in the Figure 6A is ambiguous. What is the implication for the length of each color segment in each bar to RCVE score?

Answer:

We apologize for the unclear description of Figure 6A. We added an icon for mappings between the color codes and the examined factors. The length of each color segment in each bar represents the RCVE score of the corresponding examined factor. The RCVE scores range from 0 to 1, with a higher RCVE score indicating a more important contribution of the examined factor to the model (Lines 14-16, Page 16). As shown in Figure 6A, we can find that the factor “#sample (conservation level)” generally has the largest RCVE score among the eight factors for each bar, suggesting that this factor is the most dominant determinant of circRNA reliability.

Figure 6A. The RCVE scores of the examined factors for each mock-treated sample pair. The length of each color segment in each bar represents the RCVE score of the corresponding examined factor.

13. For the figure 6C and D, What are the p-values calculated from? There were two types of measurements mentioned - numbers and correlation.

Answer:

We apologize for the unclear description of Figures 6C and 6D. All *P* values were determined using two-tailed Fisher's exact test (FET). FET was used to evaluate the difference between the percentages of not-depleted circRNAs detected in single sample and in multiple replicates.

Minor points

>> a few typing errors:

..circRNA candidates were identified by four bioinformatics tools including.. (page 7, last line) -> 'bioinformatic tools'

Answer:

We thank the reviewer for pointing out this. The typo was collected accordingly.

Reviewer 2:

In this study, the authors performed a comprehensive investigation of factors on circRNA identification reliability. Indeed, the authors concluded the importance of these factors through a generalized linear model. Briefly, the contents of the manuscript fulfills the title adequately. Here are some comments listed below:

1. The first section of results in the paper is more like an overview of the data and conditions to select the analyzed circRNA. I suggest the authors move this section to Materials and methods.

Answer:

We thank the reviewer for the valuable suggestion. We shortened this section and moved a lot of sentences to Materials and Methods. To help readers to understand the analyzed data, this newly modified section focused on how and why we filtered out potentially false positive circRNA candidates (e.g., the false positive calls due to alignment ambiguity with an alternative co-linear explanation or multiple hits) before performing the subsequent analyses of circRNA reliability. The subtitle of this section was modified as “Potential false positives in the circRNA database”.

2. Even though the statistic test are significant between non-depleted and depleted circRNAs, the differences are small. I suggest the authors to use effect size to confirm the tests' reliability.

Answer:

We thank the reviewer for the comment. All figures with the Wilcoxon rank-sum test were modified, in which the corresponding Wilcoxon effect sizes were shown. The Wilcoxon effect sizes and the corresponding 95% confidence intervals were calculated with the `wilcox_effsize` function implemented in the `rstatix` R package [6] (Lines 34-35, Page 15; Materials and Methods).

3. The authors performed a GLM analysis for the selected 8 factors to further compare their importance. I am curious about the results on performing a GLM analysis for all the factors to check if these eight factors are still significant to the circRNA identification reliability.

Answer:

We thank the reviewer for the comment. We emphasize that the GLM analysis accounts for all the eight factors simultaneously. When two or more variables (factors) are highly correlated, the effects of these factors would significantly influence each other in the model due to multi-collinearity. This would cause the insignificance of one or more factors in the model, although each single factor has been shown to be significant for assessing circRNA reliability in our results. The significance level, beta value, and

standard deviation of each single factor in the model were all provided in the supplementary data (see the “RCVE” Excel worksheet in Supplementary Table S4).

4. If the authors can make more efforts to investigate the reason why the identified factors are significant to the circRNA identification reliability, the scientific advance of this study would be largely improved.

Answer:

We thank the reviewer for the valuable suggestion. Accordingly, we added some paragraphs and rewrote the discussion section to discuss the relevance of the eight important indicators to circRNA reliability as follows. (see Line 16, Page 10-Line 31, Page 11)

“For the relative influence of each individual factor on circRNA reliability, our RCVE analysis revealed that “conservation level of circRNA” and “full-length circular sequences” were the top two dominant determinants of circRNA reliability (Fig. 6B). For the most important factor in affecting circRNA reliability, since template switching events occur randomly during reverse transcription *in vitro*, a circRNA candidate observed in multiple samples or replicates is less likely to be generated from such an *in vitro* artifact. For the second most important factor in affecting circRNA reliability, full-length circular sequences can be reconstructed by Illumina short RNA-seq reads with the reverse overlap (RO) feature [7] or identified by single-molecule long reads [8]. With the support of RO-merged reads, the BSJ events are less probable to originate from template switching because both ends of the paired-end reads are sequencing from the same fragment and reversely overlapped with each other by alignment. Similarly, with the direct support of full-length circular sequences from single-molecule long reads, the nanopore-based full-length circRNAs are less likely to be artifacts derived from template switching.

Our analyses also suggested that “supporting BSJ read count”, “supporting functional features”, and “BSJs detected by multiple tools” were good indicators for assessing circRNA reliability. Since the circularized sequence is shared with its co-linear counterpart except for the BSJ sites, the supporting BSJ read is a unique evidence of charactering a given circRNA event. It is known that most circRNAs are expressed at a low level [1, 9, 10]; particularly, a considerable number of detected circRNA candidates were supported by one BSJ read only (e.g., Fig. 1E). It is challenging to determine the reliability of such low-abundance circRNAs. Detection tools often set a threshold of supporting BSJ read count to minimize false positives. For example, circRNA_finder [11] and segemehl [12] filter circRNA candidates with a BSJ read count of at least five. In addition to BSJ reads, the functional features of the predicted

miRNA/RBP binding sites or ORFs spanning the BSJs implies the relevance of the BSJ events to functional importance. These features provide another unique support of circRNA reliability. Regarding the factor of “BSJs detected by multiple tools”, a previous study reported that combining the results of multiple tools may compensate for the possible weak points of each single tool and thereby increase the confidence of the detected circRNAs [13]. This also reflects our observation of the negative correlation between the percentages of false-positive BSJs derived from alignment ambiguity and the numbers of the detected tools (Fig. 1D).

In terms of circRNA biogenesis, we showed that the factors of “both BSJ donor and acceptor splice sites at the same co-linear transcript isoforms”, “both BSJ donor and acceptor splice sites at the annotated exon boundaries”, and “both BSJ donor and acceptor splice sites undergoing annotated AS events” were important for determining circRNA reliability. A possible reason is that circRNAs are also generated by canonical spliceosomal machinery [14-17]. Previous studies also reported that NCL transcript products (including circRNAs, fusion events, and *trans*-spliced transcripts) not matching well-annotated exon boundaries were more likely to be *in vitro* artifacts [18-20]. Particularly regarding the transcripts generated during the transcriptional process, in addition to *cis*-backsplicing, an observed intragenic NCL junction may also arise from *trans*-splicing [17, 21]. Although the majority of detected NCL events may be derived from *cis*-backsplicing, previous studies based on comparisons of poly(A)-selected and poly(A)-depleted RNA-seq data suggested that a considerable percentage of detected NCL junctions were derived from *trans*-spliced RNAs [22, 23]. Actually, a number of NCL junctions detected in human cells were experimentally validated to be generated by *trans*-splicing rather than *cis*-backsplicing [18, 21, 24, 25]. As *trans*-splicing occurs between two or more separate precursor mRNAs [26, 27], it is possible that an observed NCL junction with its two splice sites from different co-linear transcript isoforms is derived from a *trans*-splicing event. We observed that the full-length circRNAs with the support of RO-merged short reads or single-molecule long reads, which were principally not generated from *trans*-splicing, were significantly enriched for the BSJs with donor and acceptor splice sites at the same co-linear transcript isoforms (Supplemental Fig. S3).”

References

1. Guo JU, Agarwal V, Guo H, Bartel DP: **Expanded identification and characterization of mammalian circular RNAs.** *Genome Biol* 2014, **15**:409.
2. Li JH, Liu S, Zhou H, Qu LH, Yang JH: **starBase v2.0: decoding miRNA-ceRNA, miRNA-ncRNA and protein-RNA interaction networks from large-scale CLIP-**

Seq data. *Nucleic Acids Res* 2014, **42**:D92-97.

3. Xiao MS, Wilusz JE: **An improved method for circular RNA purification using RNase R that efficiently removes linear RNAs containing G-quadruplexes or structured 3' ends.** *Nucleic Acids Res* 2019, **47**:8755-8769.
4. Frees S, Menendez C, Crum M, Bagga PS: **QGRS-Conserve: a computational method for discovering evolutionarily conserved G-quadruplex motifs.** *Hum Genomics* 2014, **8**:8.
5. Menendez C, Frees S, Bagga PS: **QGRS-H Predictor: a web server for predicting homologous quadruplex forming G-rich sequence motifs in nucleotide sequences.** *Nucleic Acids Res* 2012, **40**:W96-W103.
6. Tomczak M, Tomczak E: **The need to report effect size estimates revisited. An overview of some recommended measures of effect size.** *Trends in Sport Sciences* 2014, **1**:19-25.
7. Zheng Y, Ji P, Chen S, Hou L, Zhao F: **Reconstruction of full-length circular RNAs enables isoform-level quantification.** *Genome Med* 2019, **11**:2.
8. Liu Z, Tao C, Li S, Du M, Bai Y, Hu X, Li Y, Chen J, Yang E: **circFL-seq reveals full-length circular RNAs with rolling circular reverse transcription and nanopore sequencing.** *Elife* 2021, **10**.
9. Jeck WR, Sorrentino JA, Wang K, Slevin MK, Burd CE, Liu J, Marzluff WF, Sharpless NE: **Circular RNAs are abundant, conserved, and associated with ALU repeats.** *RNA* 2013, **19**:141-157.
10. Salzman J, Chen RE, Olsen MN, Wang PL, Brown PO: **Cell-type specific features of circular RNA expression.** *PLoS Genet* 2013, **9**:e1003777.
11. Westholm JO, Miura P, Olson S, Shenker S, Joseph B, Sanfilippo P, Celniker SE, Graveley BR, Lai EC: **Genome-wide analysis of drosophila circular RNAs reveals their structural and sequence properties and age-dependent neural accumulation.** *Cell Rep* 2014, **9**:1966-1980.
12. Hoffmann S, Otto C, Doose G, Tanzer A, Langenberger D, Christ S, Kunz M, Holdt LM, Teupser D, Hackermuller J, Stadler PF: **A multi-split mapping algorithm for circular RNA, splicing, trans-splicing and fusion detection.** *Genome Biol* 2014, **15**:R34.
13. Gaffo E, Buratin A, Dal Molin A, Bortoluzzi S: **Sensitive, reliable and robust circRNA detection from RNA-seq with CirComPara2.** *Brief Bioinform* 2022, **23**.
14. Ashwal-Fluss R, Meyer M, Pamudurti NR, Ivanov A, Bartok O, Hanan M, Evantal N, Memczak S, Rajewsky N, Kadener S: **circRNA biogenesis competes with pre-mRNA splicing.** *Mol Cell* 2014, **56**:55-66.
15. Starke S, Jost I, Rossbach O, Schneider T, Schreiner S, Hung LH, Bindereif A: **Exon circularization requires canonical splice signals.** *Cell Rep* 2015, **10**:103-111.

16. Wang Y, Wang Z: **Efficient backsplicing produces translatable circular mRNAs.** *RNA* 2015, **21**:172-179.
17. Chen I, Chen CY, Chuang TJ: **Biogenesis, identification, and function of exonic circular RNAs.** *Wiley Interdiscip Rev RNA* 2015, **6**:563-579.
18. Wu CS, Yu CY, Chuang CY, Hsiao M, Kao CF, Kuo HC, Chuang TJ: **Integrative transcriptome sequencing identifies trans-splicing events with important roles in human embryonic stem cell pluripotency.** *Genome Res* 2014, **24**:25-36.
19. Kim P, Yoon S, Kim N, Lee S, Ko M, Lee H, Kang H, Kim J: **ChimerDB 2.0--a knowledgebase for fusion genes updated.** *Nucleic Acids Res* 2010, **38**:D81-85.
20. Al-Balool HH, Weber D, Liu Y, Wade M, Guleria K, Nam PL, Clayton J, Rowe W, Coxhead J, Irving J, et al: **Post-transcriptional exon shuffling events in humans can be evolutionarily conserved and abundant.** *Genome Res* 2011, **21**:1788-1799.
21. Yu CY, Liu HJ, Hung LY, Kuo HC, Chuang TJ: **Is an observed non-co-linear RNA product spliced in trans, in cis or just in vitro?** *Nucleic Acids Res* 2014, **42**:9410-9423.
22. Chuang TJ, Wu CS, Chen CY, Hung LY, Chiang TW, Yang MY: **NCLscan: accurate identification of non-co-linear transcripts (fusion, trans-splicing and circular RNA) with a good balance between sensitivity and precision.** *Nucleic Acids Res* 2016, **44**:e29.
23. Chuang TJ, Chen YJ, Chen CY, Mai TL, Wang YD, Yeh CS, Yang MY, Hsiao YT, Chang TH, Kuo TC, et al: **Integrative transcriptome sequencing reveals extensive alternative trans-splicing and cis-backsplicing in human cells.** *Nucleic Acids Res* 2018, **46**:3671-3691.
24. Takahara T, Kanazu SI, Yanagisawa S, Akanuma H: **Heterogeneous Sp1 mRNAs in human HepG2 cells include a product of homotypic trans-splicing.** *J Biol Chem* 2000, **275**:38067-38072.
25. Flouriot G, Brand H, Seraphin B, Gannon F: **Natural trans-spliced mRNAs are generated from the human estrogen receptor-alpha (hER alpha) gene.** *J Biol Chem* 2002, **277**:26244-26251.
26. Horiuchi T, Aigaki T: **Alternative trans-splicing: a novel mode of pre-mRNA processing.** *Biol Cell* 2006, **98**:135-140.
27. Gingeras TR: **Implications of chimaeric non-co-linear transcripts.** *Nature* 2009, **461**:206-211.

February 14, 2023

RE: Life Science Alliance Manuscript #LSA-2022-01793-TR

Prof. Trees-Juen Chuang
Genomics Research Center, Academia Sinica
Genomics Research Center, Academia Sinica, Taipei 11529, Taiwan
Taipei 115201
Taiwan

Dear Dr. Chuang,

Thank you for submitting your revised manuscript entitled "Assessing the impacts of various factors on circular RNA reliability". We would be happy to publish your paper in Life Science Alliance pending final revisions necessary to meet our formatting guidelines.

- please upload both your main and your supplementary figures as single files and add your supplementary figure legends to the main manuscript text
- please add the author contributions to the main manuscript text

To upload the final version of your manuscript, please log in to your account: <https://lsa.msubmit.net/cgi-bin/main.plex>. You will be guided to complete the submission of your revised manuscript and to fill in all necessary information. Please get in touch in case you do not know or remember your login name.

A. FINAL FILES:

B. MANUSCRIPT ORGANIZATION AND FORMATTING:

Sincerely,

Reviewer #1 (Comments to the Authors (Required)):

The authors have fully addressed all questions raised in the previous round of review.

Reviewer #2 (Comments to the Authors (Required)):

The authors have replied my comments well. I don't have further comments.

February 15, 2023

RE: Life Science Alliance Manuscript #LSA-2022-01793-TRR

Prof. Trees-Juen Chuang
Genomics Research Center, Academia Sinica
Genomics Research Center, Academia Sinica, Taipei 115201, Taiwan
Taipei 115201
Taiwan

Dear Dr. Chuang,

Thank you for submitting your Research Article entitled "Assessing the impacts of various factors on circular RNA reliability". It is a pleasure to let you know that your manuscript is now accepted for publication in Life Science Alliance. Congratulations on this interesting work.

DISTRIBUTION OF MATERIALS:

Again, congratulations on a very nice paper. I hope you found the review process to be constructive and are pleased with how the manuscript was handled editorially. We look forward to future exciting submissions from your lab.

Sincerely,
